# Development of an oral once-weekly drug delivery system for HIV antiretroviral therapy

Ameya R. Kirtane [1], Omar Abouzid[1,2], Daniel Minahan[1], Taylor Bensel[1], Alison L. Hill [3], Christian Selinger[4], Anna Bershteyn[4], Morgan Craig[3], Shirley S. Mo[3], Hormoz Mazdiyasni[1], Cody Cleveland [1,5], Jaimie Rogner[1], Young-Ah Lucy Lee[1], Lucas Booth[1], Farhad Javid, Sarah J. Wu[6], Tyler Grant[7], Andrew M. Bellinger[7], Boris Nikolic[8], Alison Hayward[1], Lowell Wood[4], Philip A. Eckhoff [4], Martin A. Nowak [3], Robert Langer[1,9,10] & Giovanni Traverso [1,5]

The efficacy of antiretroviral therapy is significantly compromised by medication non-adherence. Long-acting enteral systems that can ease the burden of daily adherence have not yet been developed. Here we describe an oral dosage form composed of distinct drug–polymer matrices which achieved week-long systemic drug levels of the antiretrovirals dolutegravir, rilpivirine and cabotegravir in a pig. Simulations of viral dynamics and patient adherence patterns indicate that such systems would significantly reduce therapeutic failures and epidemiological modelling suggests that using such an intervention prophylactically could avert hundreds of thousands of new HIV cases. In sum, weekly administration of long-acting antiretrovirals via a novel oral dosage form is a promising intervention to help control the HIV epidemic worldwide.

[1] Department of Chemical Engineering and David H. Koch Institute for Integrative Cancer Research, Massachusetts Institute of Technology, Cambridge, MA 02139, USA. [2] Département Biosciences, Institut National des Sciences Appliquées de Lyon, 20 Avenue Albert Einstein, Villeurbanne, France 69100. [3] Program for Evolutionary Dynamics Harvard University Cambridge, Massachusetts 02138, USA. [4] Institute for Disease Modeling, Bellevue, WA 98005, USA. [5] Division of Gastroenterology Brigham and Women's Hospital, Harvard Medical School, Boston, MA 02115, USA. [6] Department of Mechanical Engineering, Massachusetts Institute of Technology, Cambridge, MA 02139, USA. [7] Lyndra Inc, Watertown, MA 02472, USA. [8] Biomatics Capital, 1107 1st Avenue, Apartment 1305, Seattle, WA 98101, USA. [9] Media Lab, Massachusetts Institute of Technology, Cambridge, MA 02139, USA. [10] Institute for Medical Engineering and Science, Massachusetts Institute of Technology, Cambridge, MA 02139, USA. Ameya R. Kirtane and Omar Abouzid contributed equally to this work. Correspondence and requests for materials should be addressed to R.L. (email: rlanger@mit.edu) or to G.T. (email: ctraverso@bwh.harvard.edu)

Antiretrovirals have transformed disease management for human immunodeficiency virus (HIV)-infected individuals[1]. With reliable life-long adherence to combination antiretroviral therapy (ART), HIV+ individuals have a lifespan comparable to that of uninfected individuals[2]. Additionally, antiretrovirals may be taken by high-risk uninfected individuals to prevent infection, a strategy known as pre-exposure prophylaxis (PrEP)[3]. When used consistently, PrEP reduces HIV acquisition rate by 90%[4]. Despite these developments, the burden of HIV remains high worldwide. In 2015, 2.1 million people became newly infected with HIV, and there were 1.2 million HIV-related deaths[5]. These findings underscore the need to bridge the disconnect between availability of effective antiretrovirals and efficient disease control.

Lack of medication adherence to ART has emerged as a key barrier to successful HIV treatment[6,7] and prevention[8,9]. The average adherence rate to long-term ART is ~70%[7,10,11] in both high- and low-income countries, and suboptimal adherence is the strongest predictor of treatment failure and emergence of drug-resistant virus[12–15]. Further, poor adherence has also emerged as a barrier to successful implementation of PrEP[16]. For example, lack of efficacy of various formulations of tenofovir-based PrEP[17] in the VOICE trial were explained, in part, by undetectable drug levels in many participants[17] and imperfect adherence was also implicated for the lack of efficacy in the FEM-PrEP trial[18]. Additionally, analyses of the CAPRISA[19] and iPrEx trials[20] revealed that patients with better adherence to the therapeutic regimen were more likely to be protected.

Adherence levels are driven by a variety of factors, including access to affordable medications, stigma about disease status, and side effects of ART[21–23]. The high pill burden resulting from daily long-term HIV medication is known to significantly deter patient adherence[24,25]. To overcome this challenge, there has been interest in developing simplified dosage regimens. One example is single-tablet regimens, such as Atripla®, Complera®, and Stribild®, although these still require daily pill administration. Another recent breakthrough is the use of long-acting injectable nanoparticles[26–28], which release drug for weeks after intramuscular administration[29–31]. However, their long-term favourability among patients and their effect on adherence requires careful assessment, as pain and other injection site reactions have been commonly reported[32,33] and adherence to other injectables, such as hormonal contraception, is low[34,35].

We reasoned that a technology that reduces dosing frequency and is administered orally could be a promising alternative for addressing imperfect adherence to ART[36–38]. However, development of such a system has previously proven difficult due to the limited residence of drugs in the gastrointestinal tract[39]. Moreover, although structures with prolonged gastric residence have been reported[40–45], restrictions on materials that can be used for their construction limit the ability to adequately control drug release kinetics. For example, we have previously reported gastric resident dosage forms fabricated from poly(ε-caprolactone) (PCL) capable of providing both the mechanical integrity required for gastric residence and slow drug release for long-term treatment with ivermectin[42]. The coupling of mechanical resilience enabling gastric residence with drug release limited initial formulation to PCL. To develop an orally delivered dosage form with the capacity to provide extended release of a wide range of therapeutics in nearly any time release profile, a universal system capable of accommodating a wide range of polymer formulations would be required.

Here we describe such a universal system and apply it to the clinical problem of medication non-adherence in HIV. To address the challenges of prolonging gastric residence and altering drug formulations independent of mechanical properties, we describe here a modular drug delivery system which folds and recoils, enabling oral dosing, which retains its integrity in the stomach for prolonged residence, and which can be loaded with up to six different drug formulations resulting in desired pharmacokinetics. As a proof-of-principle, we show that this system is capable of delivering three highly-potent antiretrovirals—dolutegravir (DTG), cabotegravir (CAB) and rilpivirine (RPV)—for a week after a single dose in a swine model.

## Results

**Design and assembly of the gastric-resident dosage form.** The gastric resident dosage form used in these studies consists of six arms joined at a central core (Fig. 1a). The central core, made of an elastomeric material, enables folding of the dosage form into a capsule and its recoil upon dissolution of the capsule shell in the stomach. The arms provide rigidity, and capabilities to load and release the drug (Fig. 1b). To de-couple the gastric resident capacity from drug release we created a modular system in which a pocket is milled into the polymer backbone of the arms and is filled with drug–polymer matrix that controls the rate of drug release. In this paper, polymers used for the construction of the polymer backbone are referred to as structural polymers, and those used for controlling drug release are referred to as release polymers (Fig. 1b). Adopting this design enabled us to develop a gastric resident dosage form with the capacity to deliver six different drug formulations from a single structure. Significant changes could be made to the drug release kinetics and drugs with different physicochemical properties could be loaded without altering the core structure of the dosage form.

The method of assembling the dosage form is shown in Fig. 1c and Supplementary Figure 1. This design allowed us to load the system with different drug–polymer matrices, each releasing drug at a different rate. Appropriate combinations of matrices could be programmed into manufacturing to achieve desired drug release rates and plasma pharmacokinetics (Fig. 1d).

**Material selection and gastric retention.** To enable the use of multiple polymers for drug release, we sought to de-couple the two functions of the arm: rigidity for gastric residence and adjustability of drug release. To accommodate this application, construction of each part of the dosage form with optimal materials was paramount. Material selection was guided by a series of mechanical tests as described below.

A four-point bending assay (ASTM D6272 standards) was used to characterize candidate structural polymers (Fig. 2a). Two thermoplastic polyesters, PCL and poly(lactide) (PLA), showed an ultimate flexural stress of ~23.1 ± 0.9 and ~79.0 ± 4.9 MPa respectively (data is mean ± S.D., $n = 4$), and a thermoplastic polyurethane Elastollan®R6000 displayed an ultimate flexural stress of ~205 ± 14.7 MPa ($n = 4$). Tensile testing (ASTM D638 Type V standards) was used to evaluate two materials for use as elastomeric cores: a custom-made cross-linked PCL which was successfully used for the construction of a previous generation of these dosage forms[42], and a thermoplastic polyurethane Elastollan®1185, due to its ease of manufacturing. Cross-linked PCL had an elastic modulus of ~12.2 ± 3.6 MPa, while that of Elastollan®1185 was ~27.7 ± 2.2 MPa ($n = 3$) (Fig. 2b). After seven days of incubation in simulated gastric fluid (SGF), there was minimal deterioration in the mechanical properties of these materials.

The system's mechanical integrity in the stomach depended on the interface between the structural polymer and elastomeric core. We used tensile testing to determine the strength of the interface between the two materials using the same dog-bone shapes (Fig. 2c). The Elastollan®1185-Elastollan®R6000 interface showed the highest value for maximum stress (~12.1 ± 1.2 MPa) ($n = 3$).

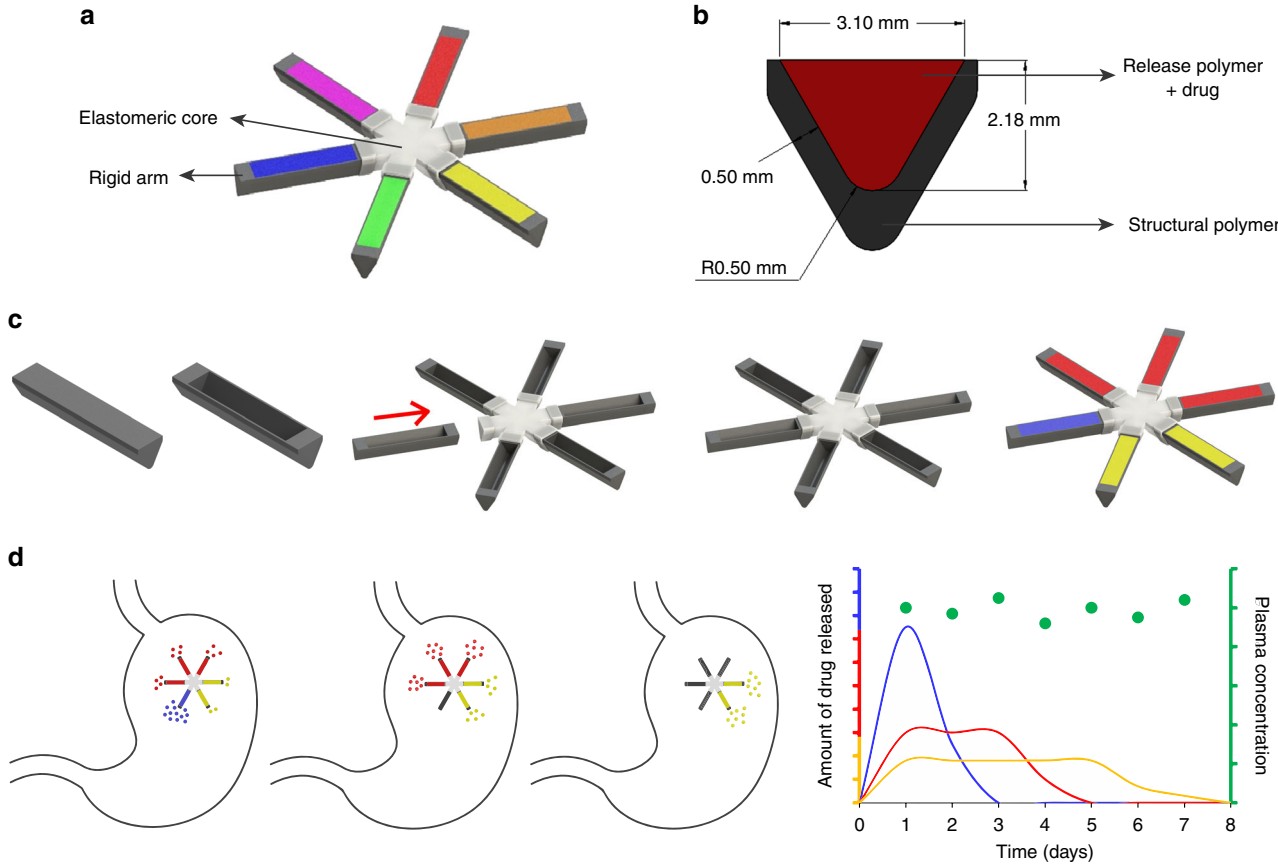

**Fig. 1** Concept of oral long acting antiretrovirals. **a** The design of the gastric resident dosage forms. The dosage form consists of an elastomeric core (grey) and six drug loaded arms (multi-coloured). **b** The cross section of the arm. The outer sleeve of the arms is made of a rigid structural polymer which provides the arm its mechanical strength. This sleeve is then filled with a drug–polymer matrix which releases the drug at a desired rate. **c** The manufacturing scheme of the dosage form. The expected performance of the dosage form in vivo is shown in **d**. The dosage form is loaded with three different polymers (blue, red and yellow) which release the drug at different rates. Selection of appropriate polymers may result in almost constant and sustained plasma drug concentrations. It should be noted that **d** is a schematic representing an ideal system, and is not experimentally obtained data

Upon administration, dosage forms experience a continuous cycling of bending forces in the stomach. To determine their resistance to these forces, we conducted 'fatigue tests' on two dosage forms (See Supplementary Figure 2a and Methods for details). The elastomeric cores of both were made of Elastollan®1185, while their arms were made of either PLA or Elastollan®R6000. The Elastollan®1185-PLA dosage forms failed in less than 2000 cycles (Supplementary Figure 2b). In contrast Elastollan®1185- Elastollan®R6000 showed no failure for up to 10,000 cycles, after which the testing was discontinued (Supplementary Figure 2c). Then, the gastric retention of these dosage forms was assessed in pigs. The results of the in vivo studies were in agreement with those of the fatigue and tensile tests. The dosage form consisting of PLA arms started to disintegrate on day 1 (Fig. 2d) and by day 4, half of the arms had separated from the core. Upon its retrieval from the animal, we found that several of the arms had broken off close to the elastomer–arm interface (Supplementary Figure 2d). In contrast, dosage forms consisting of Elastollan®R6000 arms remained intact throughout the entire study (Fig. 2d and e, and Supplementary Figs. 2e and f).

We also tested the force to pass the Elastollan®1185-Elastollan®R6000 dosage forms through a simulated pylorus using an in vitro 'funnel' test (Fig. 2f). This test was developed to provide ex vivo evaluation of prototypes to aid in the identification of optimal designs and material properties. Our previous work demonstrated that forces at or above 3 N were near optimal to maintain the dosage form in the stomach[42].

Evaluations were performed before and after week-long residence in the swine. The force required to pass the dosage forms was ~10–20 N before dosing, and decreased to ~10–15 N post-retrieval, but remained above the threshold of 3 N ($n = 10$).

We envision equipping the dosage forms with a safety mechanism that will enable disassembly in case of premature passage into the intestine. This could be a pH sensitive linker connecting the core to the peripheral arms, which is stable in the acidic environment of the stomach but dissolves in the intestine. This would facilitate break-up of the dosage form in the intestine, ensuring its safe clearance. Our group has previously developed such linkers[42,46], and here we present initial in vivo proof-of-concept. Prototypes were assembled with solid arms composed of PCL and enteric linkers connecting the arms to the elastomeric core made of the custom-made PCL elastomer described above. To evaluate the stability of the linkers and provide an approximation of an emergent clinical scenario in which an intact dosage form inadvertently passed out of the stomach, we endoscopically placed the dosage forms in the stomach and also in the duodenum. Animals were evaluated twice daily clinically and radiographically. Dosage forms deployed in the intestine started to disintegrate within 24 h, and were completely removed by six days post-administration. In contrast, all three dosage forms deployed in the stomach remained resident for six days; two of the dosage forms remained intact, and the third one only lost one arm (Supplementary Figure 3). Similar safety mechanisms can be built into structures consisting of Elastollan®R6000

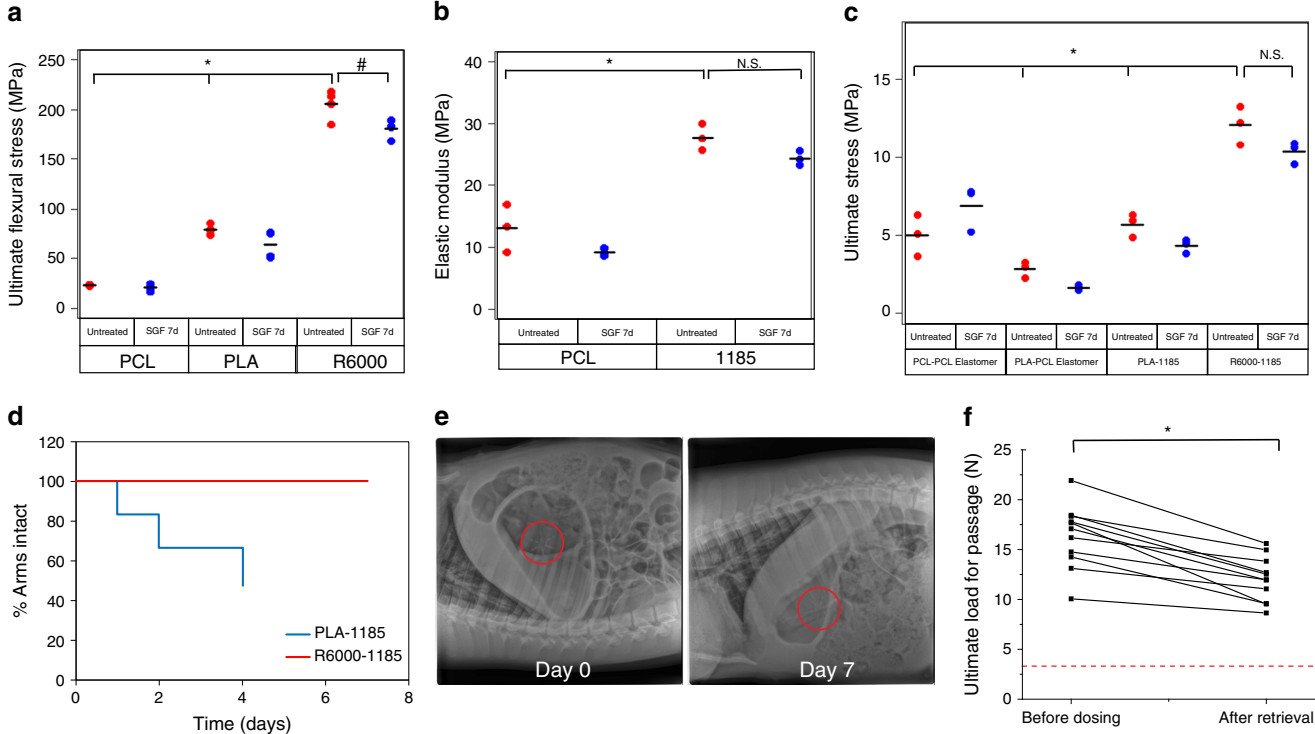

**Fig. 2** Mechanical characterization and gastric retention of gastric resident dosage forms. **a** The ultimate flexural stresses of various polymers were determined by using a 4-point bending test. Circles indicate individual measurements, and horizontal line indicates mean value. *$p < 0.05$, one-way ANOVA, post-hoc Bonferroni (comparing R6000 vs. PCL and PLA); #$p < 0.05$, two-sample $t$ test. **b** The elastic moduli of candidate central elastomers were determined using tensile testing. Circles indicate individual measurements, and horizontal line indicates mean value. * indicates $p < 0.05$, N.S. indicates not significant, two-sample $t$ test **c** The interface between candidate structural polymers and central elastomers was characterized by tensile testing. Circles indicate individual measurements, and horizontal line indicates mean value., *$p < 0.05$, one-way ANOVA, post-hoc Bonferroni (comparing R6000-1185 vs. each of the other groups); N.S. indicates not significant, two-sample $t$ test. **d** Dosage forms made of Elastollan®1185 central elastomer and either PLA or Elastollan®R6000 arms were administered to pigs. Their integrity was analysed over time. Three dosage forms containing six arms each were tested. **e** Representative X-rays of dosage forms made of Elastollan®R6000 arms and Elastollan®1185 central elastomer in a pig immediately after deployment and on day 7. **f** Funnel testing of Elastollan®R6000-Elastollan®1185 dosage forms before dosing in pigs and after retrieval from pig stomach following 7-day residence. *indicates $p < 0.05$, paired two sample $t$ test

and Elastollan®1185. For the studies discussed in this paper, linkers were excluded to isolate the effect of drug release kinetics in the gastrointestinal tract.

**Development of sustained release drug formulations**. Choice of drugs was based on the recommended daily dose of each drug (Supplementary Figure 4). Doses of DTG (an integrase inhibitor), CAB (an integrase inhibitor), RPV (a non-nucleoside reverse transcriptase inhibitor), and tenofovir alafenamide (TAF, a nucleotide reverse transcription inhibitor, and pro-drug of tenofovir) were ≤ 50 mg/day, allowing us to load sufficient mass onto the structure for weekly dosing. To further determine if these drugs were suitable for use in our system, we tested their stability at acidic pH (for gastric release) and high temperatures (for manufacturing at elevated temperatures). Unless indicated otherwise, all experiments described in this manuscript were performed with the salt forms of the drugs. DTG, CAB and RPV were stable in SGF for the duration of the study (Fig. 3a). In contrast, TAF degraded rapidly and only 10% of the drug remained stable after 10 h of incubation in SGF, leading us to discontinue further development with this drug. To assess the stability of the drugs at the elevated temperatures that can be encountered during dosage form fabrication, we heated each drug to 150 °C for 2 h. DTG, CAB and RPV were all found to be stable at these conditions (Fig. 3b); thus, we proceeded with development of dosage forms loaded with these drugs. Nuclear magnetic

resonance (NMR) and X-ray diffraction (XRD) analyses of the three drugs is shown in Supplementary Figure 5 and 6 respectively.

We loaded each of the three drugs into a variety of polymer matrices and tested their release in SGF. We chose three types of polymers—a poly(ether), poly(anhydrides), and a poly(ester)—which release drug at different rates. The release of DTG from the three types of polymers is shown in Fig. 3c. Drug release from the water soluble poly(ethylene glycol) was rapid, with ~60–70% of the drug released within the first 6 h ($n = 3$). Release from PCL was the slowest, with only 5% of the drug released over a week ($n = 3$). Total amount of drug release from the poly (anhydrides) was nearly 4–8-fold higher than that from PCL, and the rate of drug release could be altered by changing the monomer used in the synthesis of the polymer. The more hydrophilic adipic acid-based poly(anhydride) released greater amount of drug than the hydrophobic sebacic acid-based polymer. Similar trends in drug release rates were observed with RPV (Fig. 3d) and CAB (Fig. 3e).

**Pharmacokinetics of antiretrovirals**. We then analysed the pharmacokinetics of the three antiretrovirals dosed as immediate release formulations and in gastric resident dosage forms in pigs. Manufacturing of the sustained release dosage forms and the various drug formulations used are listed in Methods and Supplementary Table 1.

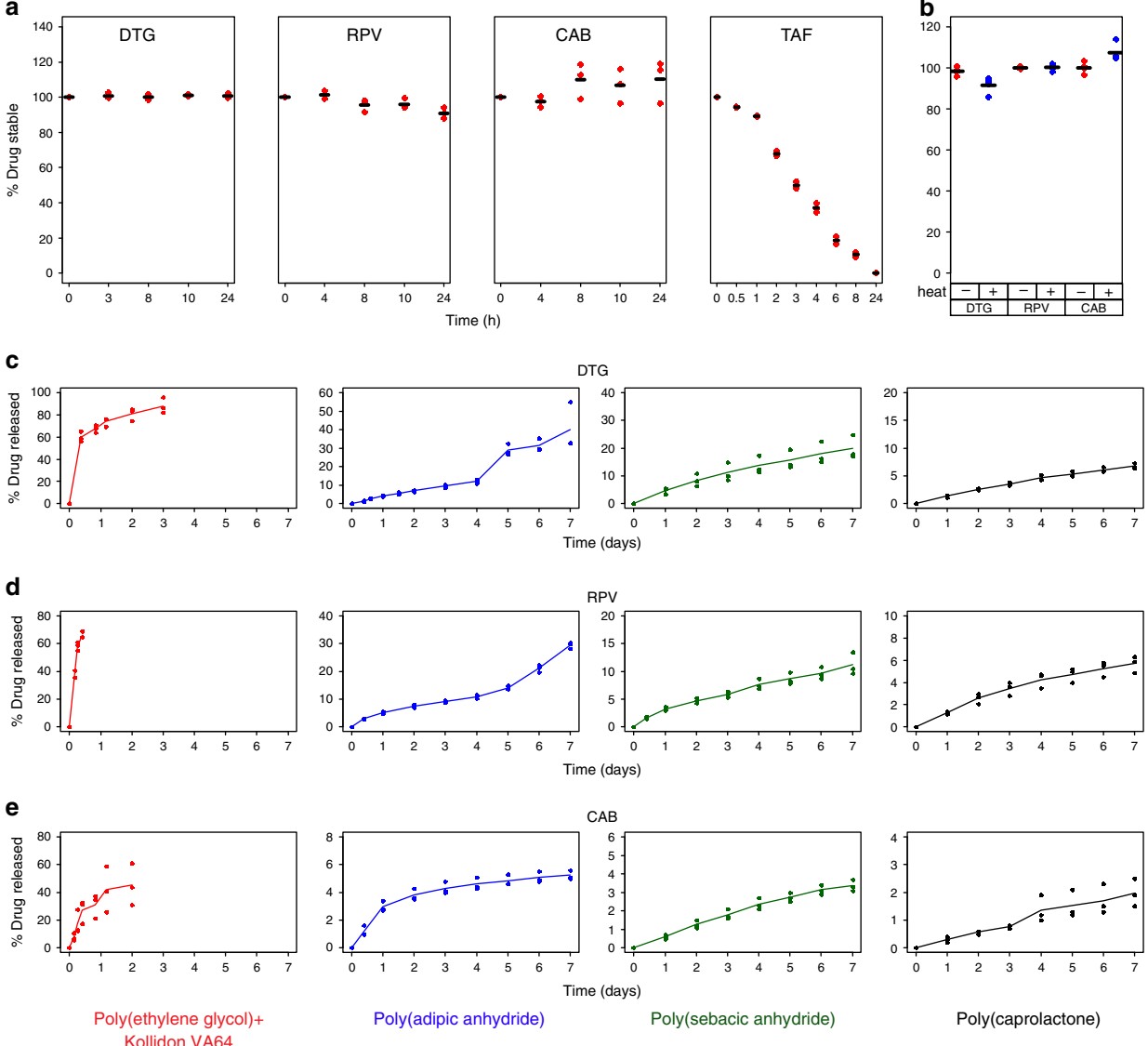

**Fig. 3** Assessment of drug stability and release rates from different polymer matrices. **a** Different antiretroviral drugs were incubated in SGF for 24 h at 37 °C. At various times, drug concentration in SGF was measured using HPLC. **b** Stability of various drugs upon incubation at 150 °C for 2 h was assessed using HPLC. Circles represent individual data points and horizontal lines indicate mean value. Release of DTG (**c**), RPV (**d**) and CAB (**e**) from various polymer matrices in SGF was measured using HPLC. Polymer matrices from left to right are: poly(ethylene glycol) + Kollidon VA64, poly(adipic anhydride), poly(sebacic anhydride) and poly(caprolactone). Circles are individual data points, and lines represent mean release profile

The plasma concentration–time profile of DTG delivered in its immediate release formulation is shown in Fig. 4a. The drug was absorbed rapidly, and detectable concentrations were observed within 15 min. An average maximum concentration of ~500 ng/mL was reached 6 h post-dose ($n = 3$). On day 1, drug concentrations were low but detectable; however, no drug was detected at or after day 2. When DTG was dosed in a gastric resident system, mean peak concentration comparable to that of the immediate release formulation was obtained within 6 h. Importantly, throughout the week following dosing, average drug concentrations of ~200–800 ng/mL were observed ($n = 3$) (Fig. 4b). We also examined drug levels in the dosage forms pre- and post-administration microscopically noting a significant void space in the arms post-administration (Supplementary Fig. 7).

RPV was also rapidly absorbed from its immediate release formulation (Fig. 4c). Drug concentrations dropped to below the limit of detection within 2 days. When dosed in a gastric resident dosage form, drug absorption was slower with maximum concentration achieved at 1 day post-dose (Fig. 4d). Drug concentrations comparable to the Cmax were maintained for the entirety of the study.

When administered as an immediate release formulation, CAB levels in the plasma peaked at 6 h (Fig. 4e), and were detectable in the plasma for up to 2 days. However, no drug levels were detected at or following day 3. When dosed in the gastric resident system, average peak plasma concentration was ~2.5-fold lower than that observed with the immediate release, but sustained concentrations (200–500 ng/mL) were maintained up to a week post-dose ($n = 3$) (Fig. 4f).

Concentration–time curves of individual animals assayed three times on the LC–MS/MS are also displayed in Supplementary Fig. 8.

**Patient level impact of long acting antiretrovirals**. Long-acting antiretrovirals will alter pharmacokinetics within treated HIV+

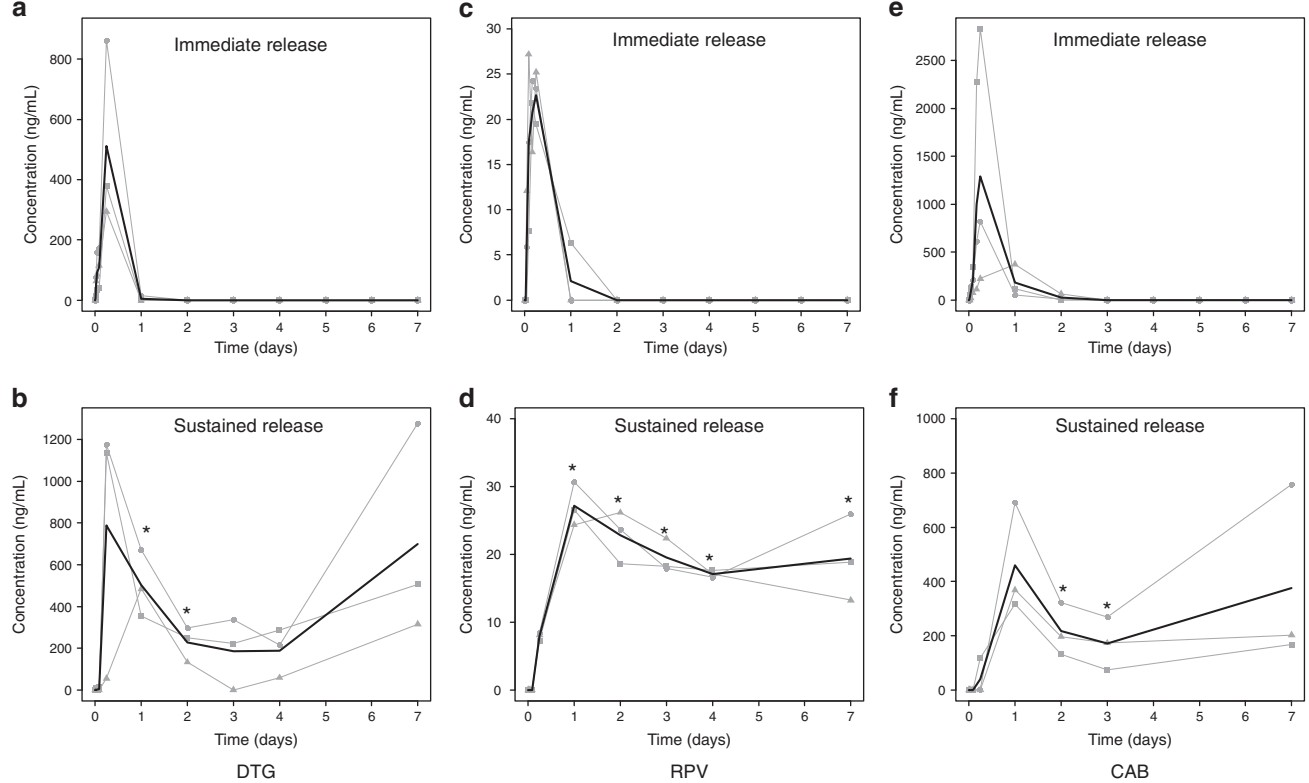

**Fig. 4** Plasma pharmacokinetics of immediate release and sustained release antiretrovirals. The concentration time profiles of (**a**) DTG immediate release (**b**), DTG sustained release (**c**), RPV immediate release (**d**), RPV sustained release (**e**), CAB immediate release and (**f**) CAB sustained release are shown. Each dosage form was tested in three animals, and plasma samples from each animal were processed three times. Data was first averaged within each animal (shown by the grey lines) and then between animals in each treatment group (shown by the black line). *indicates $p < 0.05$, two sample $t$ test comparing sustained release formulations and immediate release formulations at matching time points

individuals, and thus may also impact the degree of viral suppression. Here we adapt a previously developed framework for simulating HIV treatment outcomes to predict the impact of this new formulation[47]. Our model combined patient adherence behaviour, pharmacokinetic profiles, pharmacodynamic functions, and a multi-strain version of the well-established viral dynamic model that tracks mutation and selection between drug-resistant and sensitive strains (see Supplementary Notes 1). As a baseline case, we assumed that weekly long-acting therapy has the same peak and trough drug levels as existing formulations, but 7-fold longer half-life (Fig. 5a).

Models predict that DTG and RPV given as maintenance monotherapy are similarly effective in their long-acting form versus the standard, daily formulation for a range of adherence levels (Fig. 5b, c, d, f and g). The fraction of treatment failures accompanied by emergence of drug resistance increases with the long-acting form for RPV but not DTG. In the conservative scenario of "weekly-choice adherence", we assumed that individuals who had, for example, a 70% chance of taking any given pill of their standard, daily regimen would also decide with 70% likelihood, at the start of each week, to either take their long-acting formulation or skip it for the week. In the more optimistic "daily-choice adherence" scenario, we assumed these individuals had a 70% chance of taking the next long-acting dose each day after their previous dose period has ended. The latter scenario resulted in a lower chance of individuals going for long periods without drug coverage, and is predicted to significantly improve patient outcomes with long-acting antiretrovirals (~6-fold reduction in failures), even if drug mass in each dose is reduced (Fig. 5e and h).

**Population level impact of long acting antiretrovirals**. A particularly promising application for long-acting antiretrovirals is for PrEP in high-risk uninfected individuals. To gauge whether such a use case could make an impact on combating the spread of HIV/AIDS in high-burden countries, we used modelling to estimate how many new infections could be prevented with the use of a long-acting oral antiretroviral dosage form for PrEP, as compared to currently available daily oral PrEP. We started by reviewing a recent meta-analysis[48] which estimated the difference in adherence to daily versus weekly dosage forms of similar medications prescribed for the same condition. While the original meta-analysis reported odds ratios for high adherence (defined as medication possession rate or proportion of days covered >80%, see Supplementary Methods), we repeated the analysis to estimate the absolute difference, yielding an estimated increase of 15% (95% CI: 14–17%) in the highly adherent population faction (Supplementary Fig. 9) when switching from daily to weekly dosing. Efficacy of PrEP for HIV prevention is strongly correlated to the fraction of recipients having detectable plasma drug concentrations, which in turn is related to adherence. In the Partners PrEP study, 84% of highly adherent individuals had detectable levels of tenofovir in their blood. Regression analysis across oral PrEP trials yielded an estimated 1.54% increase in efficacy, defined as the reduction in per-exposure probability of infection, for every 1% increase in fraction with detectable drug levels. Combining evidence about the influence of a weekly regimen on adherence[48], the relationship between adherence and detectable drug levels[49], and the relationship between detectable drug levels and PrEP efficacy[50,51] (see Supplementary Methods), we arrived at a plausible range of 20% (95% CI: 18–22%) increase in efficacy for weekly PrEP as compared to daily PrEP in

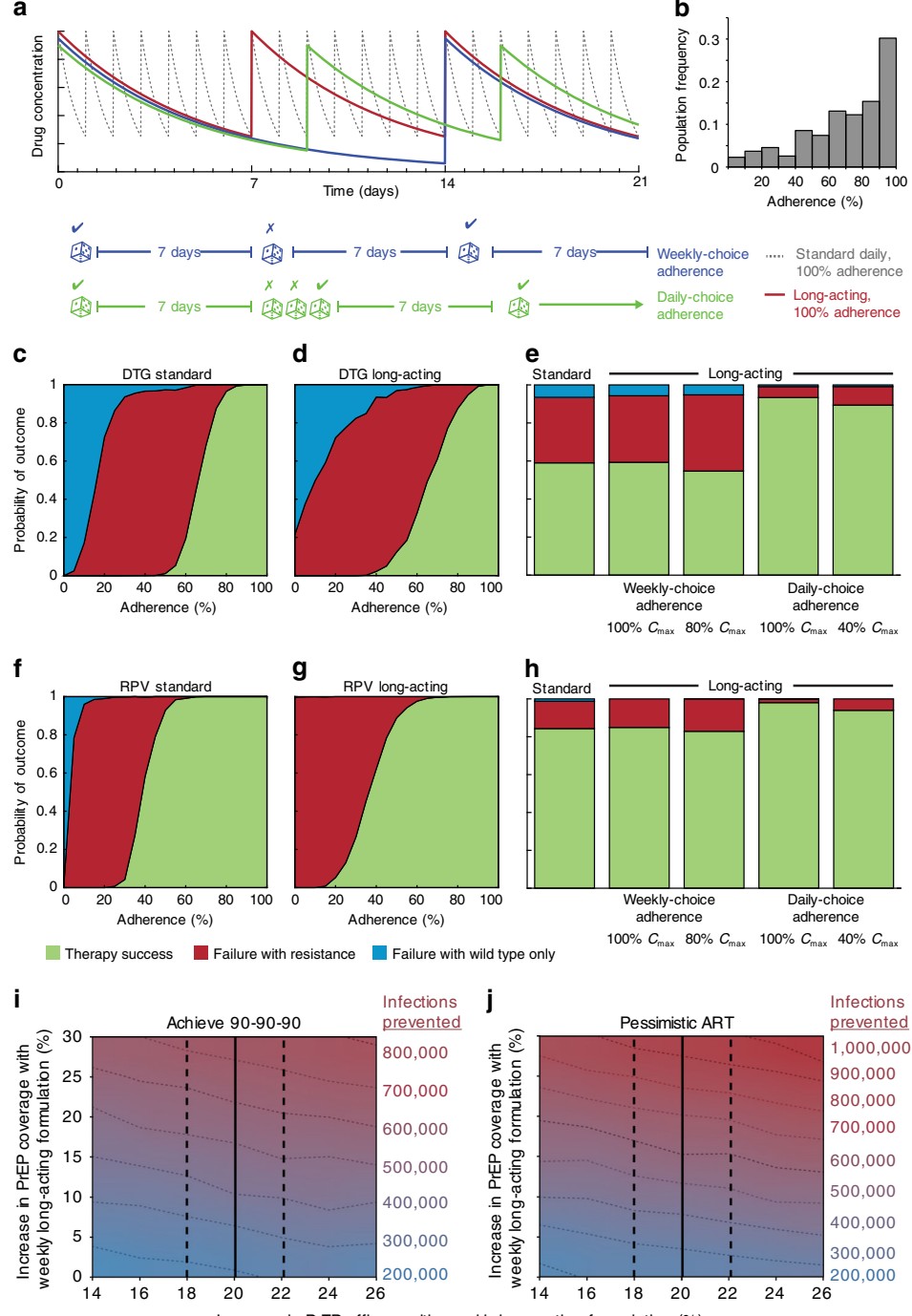

**Fig. 5** Mathematical modelling of HIV treatment outcomes using long-acting antiretrovirals. **a–h** Viral dynamics modelling of HIV growth and evolution within individual patients. We consider a hypothetical weekly formulation, with identical peak concentration and 7-fold longer half-life **a**. Adherence patterns described in more detail in text and Supplementary Information. Patients are initially fully virologically-suppressed and failure is defined as two consecutive monthly viral load measurements > 200 c/mL in the year-long follow-up period. The height of the shaded area indicates the probability of outcome. Predicted patient outcomes versus adherence level (% of scheduled doses taken) for existing daily formulation of DTG (**c**) and RPV (**f**), and for long-acting versions, (**d**, **g**). Predicted outcomes for DTG (**e**) and RPV (**h**): a cohort of patients with a distribution of adherence levels taken from empirical data[11], average adherence ~70% (**b**). $C_{max}$ is the peak drug level and is determined by the dosage size and absorption. 100% means that this value is the same for the hypothetical long-acting formulation as the existing one. **i–j** Epidemiological modelling of the impact of using oral long-acting antiretrovirals for PrEP in South Africa. The contours show the cumulative number of additional infections prevented (in thousands) by switching from daily to weekly oral PrEP formulation in South Africa by 2037. The *x*-axis depicts the supposed increase in efficacy (defined as reduction in per-exposure probability of infection), compared to a baseline efficacy of 50%. Solid lines indicating the plausible estimates (confidence intervals depicted as dashed lines) from meta-analyses respectively. The *y*-axis shows assumed increase in coverage (% of high-risk population receiving PrEP) assuming a baseline coverage of 30%. "Pessimistic ART" assumes current rates of ART use for treatment (**i**) while "Achieve 90-90-90" assumes increased rates (**j**)

high-risk populations (blue lines in Fig. 5i and j). However, because weekly PrEP will likely involve different drugs and may influence efficacy and coverage in unexpected ways, a wider range of efficacy and coverage assumptions were input into the EMOD v2.5 micro-simulation model[52–55], a network transmission model of HIV that includes heterosexual and mother-to-child HIV transmission, calibrated to the HIV epidemic in South Africa[56–58]. Assuming that availability of a weekly regimen influences only efficacy and not coverage, the model predicts approximately 200,000 additional infections averted over 20 years—a 3.4% cumulative reduction—by making a weekly PrEP formulation available for high risk populations ages between 15 and 29. Contours (Fig. 5i and j) show the variability of these estimates under different assumptions about the increment in efficacy (x-axis), the possibility of increasing PrEP coverage due to greater acceptability of a weekly regimen (y-axis), and assumptions whether treatment scale-up could be achieved (left and right panels). For example, if PrEP coverage were to increase from 30 to 60% as a result of the availability of a weekly PrEP regimen, the number of additional infections averted would increase to between 700,000 and 900,000 (assuming scale-up of HIV treatment to achieve the 90-90-90 goals or maintenance of present-day treatment coverage, respectively).

## Discussion

Engineering- and formulation-based approaches are at the forefront of efforts directed towards simplifying the antiretroviral dosing regimen[59]. To that end, we have developed an orally-administered platform that can deliver sustained plasma drug concentrations of three antiretrovirals (DTG, CAB and RPV) for up to seven days after a single administration event. Several other long acting drug delivery systems are currently under investigation[59], with vaginal rings[60], oral[61] and injectable nanoparticles[62] being at the most advanced stages of clinical testing. Treatment with injectable nanoparticles of CAB and RPV, and vaginal rings of dapivirine have resulted in sustained systemic and local concentrations respectively, while orally administered nanoparticles of lopinavir and efavirenz have afforded enhanced bioavailability and consequent reduction in dose. However, the approach presented here is distinct from other strategies in its non-invasiveness and its ability to achieve long-lasting drug levels in systemic circulation.

The choice of drugs in the current work was motivated in part by the mounting evidence supporting the efficacy of DTG, CAB and RPV either alone or in combination (integrase inhibitor +non-nucleoside reverse transcriptase inhibitor) for maintenance therapy. For example, the combination of CAB and RPV was recently tested in HIV-1 patients who had achieved viral suppression[63]. In these patients the combination was found to be safe and effective at maintaining viral suppression for up to 72 weeks. Other clinical trials are underway to evaluate safety and tolerability of a combination of long acting formulations of CAB and RPV in healthy individuals (clinicaltrials.gov identifier: NCT01593046) and their efficacy in patients (clinicaltrials.gov identifier: NCT02938520) that have achieved viral suppression with an integrase inhibitor-based single tablet regimen. DTG has also been shown to be effective as monotherapy or in combination with RPV for treating patients with suppressed viral loads[64–68], albeit cases of resistance to DTG monotherapy are emerging and its use as such needs serious consideration[69–71]. In addition to these clinical studies, a long acting formulation of CAB has been shown to protect against simian immunodeficiency virus infection in macaques[29,30], providing support for its clinical testing for PrEP. We do not anticipate all three drugs to be used together in a single patient.

To evaluate the potential clinical and public health impact of long-acting oral antiretrovirals, we employed established mathematical modelling frameworks to predict their impact on patient outcomes and epidemiologic trends. We first examined the impact of long-acting DTG and RPV in the context of treating existing infection. We evaluated each drug individually, and assumed that patients who were completely virologically suppressed by existing combination therapy were transitioned to long-acting maintenance monotherapy. Predictions for long-acting DTG and RPV were favourable, especially if individuals who missed scheduled doses could decide to restart any day of the week. In reality, drugs may be given in combinations, and may be used in treatment-naïve patients, altering these predictions. Adherence patterns may change in more complex ways than simulated here. Like any model, ours is limited by the availability of detailed data characterizing pharmacokinetics and pharmacodynamics, knowledge of dominant resistance pathways, and the reliability of ex vivo/in vivo correlations in viral fitness. The limited data on CAB, which is not yet FDA-approved, precluded its inclusion in this modelling.

Evidence from other medications prescribed for long-term use, and for which both daily and weekly oral dosage forms are available, implies that adherence to weekly formulations is substantially higher. We evaluated the potential impact of sustained-release antiretrovirals used prophylactically on the spread of HIV in South Africa, an archetypal high-prevalence setting. Mathematical modelling confirms that the improvements in adherence typically seen with weekly dosing have the potential to avoid hundreds of thousands of new HIV infections. The population-level impact of this technology would be greatly magnified if weekly dosage forms are also proven to be more acceptable to end users, enabling administration to a larger numbers of individuals. This analysis was performed using efficacy and adherence data from trials of tenofovir disoproxil fumarate for PrEP. An implicit assumption of our modelling is that any new drug that would be approved for PrEP use in the general population would be at least as effective as current tenofovir disoproxil fumarate-based prevention. However, to account for potential differences, we consider a wide range of efficacy values in our sensitivity analysis. Once the pharmacology and efficacy of the drugs we used in our studies is established, such a framework can be applied to those drugs as well.

There are several limitations to the current study that need to be addressed going forward. We used a swine model for studying the gastric residence and antiretroviral pharmacokinetics, as the anatomy of the pig stomach closely resembles that of humans[72] and pigs used in this study had a body weight comparable to adult humans. However, gastrointestinal transit time in swine is slower than in humans[73]. For instance, in humans fed with indigestible pellets, 50% of the feed is emptied from the stomach in ~5 h[74], while in piglets fed with pellets of similar density, it can take up to 9 h for 50% emptying[73]. Moreover, the gastric emptying rate can depend significantly on the age and size of the animal, and whether it is in a fasted or fed state. The pigs used in our study were fed a broad diet, and we did not observe any significant anomaly in dosage form performance associated with a particular component of the diet. However, it is possible that certain diets may impact the gastric residence of our dosage form, and understanding these effects will be key for future development.

Another limitation of using the pig model is that it precludes evaluating the dosage form in the context of HIV disease. Additionally, the plasma concentrations of the antiretrovirals observed in our study were lower than those achieved in clinical studies involving these drugs, which may be due to physiological differences between pigs and humans. For example, in pigs the intrinsic clearance and volume of distribution of drugs such as prednisolone and cyclosporine A is higher than those in humans[75]. To address these issues, further testing in other large mammals including non-human primates will be necessary.

Differences between the inactive ingredients used in the commercial formulations and ours could have also contributed to these differences in pharmacokinetics.

In animals dosed with DTG sustained release dosage form, there was an upswing in the concentration observed on day 7, which is likely due to the slowest drug releasing matrix loaded onto the dosage form [i.e. the poly(sebacic anhydride)-based matrix]. Removal of this formulation from the dosage form may result in declining concentrations throughout the seven day period. Further studies in a disease-relevant animal model (such as non-human primates) and better understanding of the drug pharmacokinetic-pharmacodynamic properties can help guide future formulation optimization. An in-depth evaluation of pharmacokinetic-pharmacodyamic properties of these drugs will also inform best practices in scenarios where the dosage form or a part of it prematurely passes into the intestine. There is also considerable inter-animal variability in the plasma drug concentrations of the antiretrovirals. This may be because the polymer drug matrices were manufactured by hand and the variabilities can potentially be reduced by scalable manufacturing methods.

There are currently some restrictions on drugs that can be delivered using this platform. During manufacturing and following delivery, drugs are exposed to elevated temperatures, acidic pH and high humidity making it challenging to deliver drugs that are unstable under such conditions. Additionally, in its current state the dosage form can support ~10–30% by weight of the drug. As a result, only drugs that have a relatively low daily dose (~50 mg or lower) can be effectively administered. As highlighted in Supplementary Figure 4, only four drugs have a dose low enough to be effectively administered using our system, and of these TAF due to its instability in acidic pH could not be used with our system. The remaining drugs, DTG, RPV, and CAB —still an investigational drug—are all being considered for dual[32,33,63,66] or monotherapy[29,62,64,65,76] maintenance or prophylaxis, including in combination with each other and models that incorporate existing knowledge of pharmacometrics and resistance pathways suggest favourable outcomes. Therefore while the introduction of more low-dose HIV drugs will intensify the impact of this technology, it can potentially be developed even with existing antiretrovirals.

Ultimately, control of the worldwide HIV epidemic requires innovations to increase the uptake of highly-active ART. We show here the development of the first orally-available sustained-release antiretrovirals. This system can currently deliver sustained concentrations of DTG, CAB and RPV for up to 1 week. Clinical translation of this strategy would represent a paradigm shift in HIV treatment and prevention.

## Methods

**Materials**. Poly(lactide) (PLA) was purchased from NatureWorks LLC, USA. All Elastollan® polymers were obtained from BASF. Dolutegravir (DTG), cabotegravir (CAB), tenofovir alafenamide and rilpivirine (RPV) were purchased from Hangzhou Hysen Pharma Co. Ltd., China. All other chemicals were purchased from Sigma Aldrich, USA. Poly(dimethyl siloxane) (PDMS) negative moulds for manufacturing dosage forms and bars for mechanical testing were purchased from Proto Labs, USA.

**Flexural strength of polymers used as structural polymers**. Polymers were selected for use as the structural polymer in the dosage form based on a four-point bending flexural test. The test was performed on an Instron® 5943 and a servo-hydraulic 1331 (Instron®, USA) according to ASTM D6272 standards. A 500 N load cell was used for bars made out of poly(caprolactone) (PCL) and PLA, and a 5 kN load cell was used for bars made out of Elastollan®R6000.

Bars (ASTM D6272 standards) consisting of PCL (Mn ~45 KDa) and PLA were prepared by melting the polymers in negative moulds at ~100 °C and 220 °C respectively, followed by cooling at room temperature. Elastollan®R6000 bars were prepared by extruding the polymer and then injection moulding it in a custom-made stainless steel die. Extrusion was carried out using an Xplore™ twin screw microcompounder (Xplore™ Instruments, Netherlands) at 235 °C and 75 r.p.m.

Injection moulding was carried out on an Xplore™ 5.5 cm³ laboratory injection moulding machine. The injection moulder barrel and plunger were heated to 240 °C and the mould temperature was set at 22 °C. The material was injected into the mould using a ramp function that controlled the pressure. It started at 4 bar for 1 s, ramped up to 5.5 bar over 1 s, and was held at 5.5 bar for an additional 5 s.

**Elastic modulus of polymers used as central elastomers**. The suitability of materials for use as elastomeric cores in the dosage form was determined, in part, by measuring their elastic moduli. To obtain the elastic modulus of each material, tensile testing was performed as described in the ASTM D638 Type V test.

For the test, Elastollan®1185 dog bones were manufactured by melting the polymers in negative moulds at 245 °C and then cooling at room temperature. A custom-made elastomeric PCL was also tested[42]. The elastomeric PCL was prepared as described previously[42] by mixing linear PCL (Mn ~45 KDa) with hexamethylene diisocyanate-mediated cross-linked PCL diol (Mn ~530 Da) and PCL triol (Mn ~900 Da).

**Interface between central elastomer and structural polymer**. The interface between the central elastomer and the structural polymer was evaluated by tensile testing. Tensile testing was performed as described for the central elastomers in the previous section. In these experiments, one-half of the dog bones was made with a candidate elastomeric polymer, and the other half was made with a candidate structural polymer. The two polymers were interfaced at a temperature above the melting point of at least one of the polymers for ~30 min, before cooling them at room temperature.

**Assessing drug stability**. All drugs were converted to their respective salt forms as described in Supplementary Methods.

To determine drug stability at low pH, the drugs were dissolved in methanol to form a concentrated stock solution. Methanol solutions were then diluted 20-fold in simulated gastric fluid, USP without pepsin (pH ~1.2; henceforth referred to as SGF). Drug solutions were then placed in a shaker incubator (New Brunswick™ Innova® 44, Eppendorf, USA) at 37 °C and 150 r.p.m. At various times, aliquots were withdrawn and stored at −20 °C until analysis. Drug concentrations in the solution were analysed using HPLC and reported as percent of the initial concentrations.

HPLC was performed on an Agilent 1260 Infinity HPLC system (Agilent Technologies Inc., USA) equipped with Model 1260 quaternary pump, Model 1260 Hip ALS autosampler, Model 1290 thermostat, Model 1260 TCC control module, and Model 1260 diode array detector. The output signal was monitored and processed with ChemStation® software (Agilent Technologies Inc., USA). Chromatographic separation was carried out on a 50 mm × 4.6 mm EC-C18 Agilent Poroshell 120 analytical column with 2.7 μm spherical particles.

For DTG, the column was maintained at 40 °C and the mobile phase consisted of acetonitrile: water (pH 3.5 adjusted with 0.1% formic acid) (25:75 v/v) at a flow rate of 1.2 mL/min over a 12 min run time. The injection volume was 5 μL, and the UV detection wavelength of 260 nm was selected.

For RPV, the stationary phase was maintained at 40 °C. The mobile phase consisted of methanol: water (pH 3.5 adjusted with 0.1% formic acid) (43:57 v/v) at a flow rate of 0.9 mL/min over a 10 min run time. The injection volume was 20 μL, and the UV detector was set at 290 nm.

For CAB, the column was maintained at 45 °C and the optimized mobile phase consisted of acetonitrile: water (pH 3.5 adjusted with 0.1% formic acid) (30:70 v/v) at flow rate of 0.8 mL/min over an 8 min run time. For analysis, 5 μL of the sample was injected on to the column, and the samples were analysed at a wavelength of 260 nm.

For tenofovir alafenamide, the column was maintained at 22 °C. The mobile phase consisted initially of acetonitrile: water (pH 3.5 adjusted with 0.1% formic acid) (100:0 v/v). It was ramped up to acetonitrile: water (pH 3.5 adjusted with 0.1% formic acid) (0:100 v/v) over 3.5 min and held at this ratio for 1 min. At 4.5 min, the mobile phase was brought back to acetonitrile: water (pH 3.5 adjusted with 0.1% formic acid) (100:0 v/v) over 1 min, and held there until the completion of the run. The flow rate was 0.6 mL/min and the run time was 8 min. The injection volume was 10 μL, and the UV detection wavelength of 260 nm was selected.

To determine the drug stability at high temperatures, the drug was weighed in a 20 mL glass vial, and placed in a convection oven at 150 °C for 2 h. Drug was dissolved in methanol, and the concentration was compared to that of an unheated standard of the same initial concentration via HPLC.

**Synthesis and characterization of polyanhydrides**. Poly(sebacic anhydride) was synthesized by a two-step reaction as described previously[77]. Briefly, a prepolymer was prepared by refluxing sebacic acid with a 10-fold molar excess of acetic anhydride for 20 min at 140 °C. The unreacted acetic anhydride was then removed under reduced pressure at 50 °C using a rotavap. The remaining product was dissolved in toluene and precipitated in ether overnight to yield the prepolymer. The product was filtered and dried overnight. The prepolymer was then melt-polymerized at 180 °C under vacuum (~1 mbar) for ~90 min. Every 30 min, the reaction mixture was stirred and subjected to a strong nitrogen sweep (for ~30 s). The reaction mixture was cooled and dissolved in dichloromethane. The polymer was precipitated in ether, filtered and dried overnight before further use.

Poly(adipic anhydride) was synthesized as described by Dinarvand et al[78]. Adipic acid was reacted with 10-fold molar excess of acetic anhydride under reflux at 140 °C

for 4 h. Acetic anhydride was removed under vacuum at 50 °C using a rotavap. The mixture was then dissolved in toluene and refluxed at 110 °C for 2 h. Using a rotavap, the solvent was removed under vacuum at 50 °C. The white product was dissolved in toluene and precipitated in ether overnight to yield the prepolymer. The product was collected by filtration and dried under vacuum. In the next step, the prepolymer was dissolved in dichloromethane and polymerized at room temperature for 1 h using triethylamine (0.4 mol% of the prepolymer), as the initiator. The polymer was then precipitated in ether, filtered and dried overnight before further use.

The identity of the polymers was confirmed using $^1$H NMR (Avance 400 MHz spectrometer, Bruker®, USA). The molecular weight of the polymers was determined using an Agilent® GPC coupled with a triple detector array 305 (Malvern, USA). A LT6000L mixed high organic column (Malvern, USA) was used as a stationary phase. The weight average molecular weight of poly(sebacic anhydride) ranged 9–20 kDa and that of poly(adipic anhydride) ranged 2–5 kDa.

**Synthesis and characterization of drug polymer matrices.** Drug loaded polymer matrices were synthesized by melt mixing. Appropriate amounts of drug and polymer were weighed and mixed in a 20 mL glass vial. The vial was then placed in a convection oven at 130 °C to melt the polymer. The mixture was then transferred to a negative mould heated to the same temperature. Once the mould was filled with the melted mixture, it was removed from the oven and cooled at room temperature.

Drug release studies were performed in SGF supplemented with 5%w/w Tween® 20. The drug–polymer matrices were placed in 50 mL release media at 37 °C in an incubator shaker at 150 r.p.m. At various times, about 1 mL of the media was collected and stored at −20 °C until further analysis. The rest of the media was discarded and the drug–polymer matrix was placed in fresh media. Drug concentrations in the aliquots were measured using HPLC as described previously.

**Manufacturing of gastric resident dosage forms.** Based on the in vitro studies, we selected polymers for the manufacturing of dosage forms. The elastomeric core in the dosage forms used in our studies was made of Elastollan®1185. Elastollan®R6000 was used as the structural polymer, and a combination of poly(sebacic anhydride), poly(adipic anhydride) and poly(ethylene glycol) as the release polymers.

We first manufactured the Elastollan®R6000 arms using a microcompounder extruder and injection moulder as described above. Post-moulding, pockets were milled into these arms using an Othermill CNC (Other Machine Co., USA) using a file generated in MasterCAM software (Mastercam, USA). The arms were secured in a custom 3D printed fixture in order to allow for repeatable CNC machining. We developed the capability to machine multiple arms in one run to scale up manufacturing. The feed-rate of the machine was 75 mm/min with a spindle speed of 10,500 r.p.m.

The Elastollan®R6000 arms were then placed in the negative moulds, and Elastollan®1185 was melted in the central part of the mould at 245 °C. Once the Elastollan®1185 melted, the Elastollan®R6000 arms were slowly inserted in the central elastomer (about 5 mm) such that they were over-moulded in it. The dosage forms thus formed were removed from the oven and allowed to cool at room temperature.

Release polymers [poly(ethylene glycol)-Kollidon VA64 (5:1), poly(sebacic anhydride) or poly(adipic anhydride)] were mixed with drug (40%w/w drug), and were heated to 130 °C as described before. Different amounts of these mixtures were transferred to the pocket in the Elastollan®R6000 arms in the dosage form at 130 °C. The relative amounts of mixture used in the dosage forms are listed in Supplementary Table 1. Once the dosage form was filled with the required amounts of drug–polymer matrices, it was removed from the oven and allowed to cool at room temperature. This procedure is shown in Supplementary Figure 1.

To make dosage forms with pH sensitive linkers, enteric films were solvent cast by dissolving an enteric copolymer, L100-55 (Evonik), and an adhesive plasticizer, Plastoid B (Evonik), in acetone at a 90:10 ratio. Following evaporation of the solvent a flexible film 500 μm in thickness was formed. The film was cut into 3 mm × 3 mm squares and solvent welded to arms of the dosage form around break points using 50 μL of acetone. Dosage forms were stored at 23 °C for 24 h prior to dosing in animals. In the current paper, pH sensitive linkers were introduced in dosage forms made of PCL, however with further optimization, similar linkers can potentially be introduced in dosage forms made from other materials as well. We also anticipate that such dosage forms can be produced at scale using manufacturing techniques such as injection moulding and extrusion.

**Evaluation of fatigue testing of dosage forms.** We tested two types of dosage forms under cyclic loading using a custom-made funnel-shape setup to simulate folding/unfolding of the structure[42]. Both types were made of an Elastollan®1185 central elastomer; one group had PLA arms while the other one had Elastollan®R6000 arms. As shown in Supplementary Figure 2a, the structure was placed on top of a funnel shape on a uniaxial mechanical tester (Instron 5943, Norwood, MA) and attached to a stainless steel rod using single thread string and Krazy® glue around and on the centre of the dosage form respectively. This rod pushed the dosage form through the funnel shape while constraining the structure to minimize noise during the test. To achieve the most destructive case, the structure was pressed in the direction opposite to its folding direction. A trapezoidal cyclic displacement with $d_{max} = 12.94$ mm was applied to the structure: (i) pushed down to the funnel to $d_{max}$ in 5 mm/s speed, (ii) kept at $d_{max}$ for 0.5 s, (iii) returned to its un-deformed shape at 0.5 mm/s and (iv) kept at its un-deformed shape for 0.5 s. $d_{max} = 12.94$

mm was chosen so that the span of the structure was at ~25 mm in its folded shape. The cut-off of 25 mm was chosen based on previous estimates of the size of the human pylorus[79]. Three samples of each dosage forms were tested for fatigue.

**Funnel testing of gastric retentive dosage forms.** In order to evaluate the force required to pass an intact dosage form through the pylorus, a custom funnel test was used as described above. Using an Instron machine, the dosage form was tested in an apparatus consisting of a fixed funnel and a piston. The dosage form was rested inside the opening of the funnel in the orientation that allows for folding, while all arms made contact with the funnel walls. The piston was then lowered causing the dosage form to be folded along the contours of the funnel walls until it reached the smaller diameter of 20 mm at the bottom of the funnel. The peak force was recorded after 45 mm of displacement at a rate of 60 mm/min from initial contact of the piston and dosage form.

**Evaluation of pharmacokinetics of antiretrovirals.** All animal experiments were approved by the Committee on Animal Care at the Massachusetts Institute of Technology. In vivo pharmacokinetics were evaluated in an unblinded fashion in female Yorkshire pigs (50–95 kg) that were randomly assigned to the immediate release or sustained release groups. Animals were fed daily in the morning and in the evening with a diet consisting of pellets (Laboratory mini-pig growler diet, 5081), with a midday snack consisting of various fruits and vegetables. Specifically, the pellets were 5/32" x ¼" length, and consisted of ground oats, alfalfa meal, wheat middlings, soybean meal, dried beet pulp, salts and other micronutrients.

To determine the oral pharmacokinetics of immediate release dosage forms of DTG and CAB, 50 mg of DTG and 30 mg of CAB were weighed and filled into gelatine capsules. For RPV pharmacokinetics, RPV base was dissolved in poly (ethylene glycol) (molecular weight—400 Da) containing 10-fold molar excess of citric acid. The concentration of RPV base in this solution was 5 mg/mL and 5 mL of the solution was dosed in a gelatine capsule. The solution was filled into the capsule immediately prior to dosing, as a result the capsule did not soften before administration. To determine oral pharmacokinetics from gastric retentive dosage forms, the arms of dosage forms (2–3/pig containing 350 mg DTG, 175 mg RPV or 210 mg CAB) were folded into capsule shells immediately prior to dosing. At the time of dosing, the pigs were sedated using a cocktail of Telazol® (5 mg/kg), xylazine (2 mg/kg) and atropine (0.04 mg/kg). The capsules containing drug or dosage form were delivered into the intestine or stomach respectively. At various time points, blood samples (0.5 mL) were collected from a central line connected to the external jugular vein or from a mammary vein into BioLegend® Vacutainer gold top tubes (Becton, Dickinson and Co. USA). Plasma samples were separated from blood by centrifugation (1800g, 10 min at 4 °C) and were stored at −80 °C. Atropine can reduce gastric acid production and gastric motility which may affect the overall drug pharmacokinetics and the gastric retention of the dosage form. While information about interaction between atropine and CAB is not yet available, no interactions between atropine and DTG or RPV have been reported. Additionally, the gastric retention of and drug release from the gastric retentive dosage forms were evaluated for a week, while the terminal half-life of atropine is ~2 h[80]. Hence we believe the use of atropine has, if at all, a minor effect on the overall findings of our study.

Drug concentrations in the plasma were analysed using UPLC-MS/MS. Plasma samples (100 μL) were spiked with 200 μL acetonitrile solution of 250 μg/mL maraviroc (internal standard) to cause precipitation. Samples were vortexed and sonicated for 10–20 min and then centrifuged (21130g, 10 min). Two hundred microliters of solution were pipetted into a 96-well plate containing 200 μL of nanopure water. Finally, 2.5 μL was injected onto the UPLC–ESI–MS system for analysis.

UPLC separation was carried out on a Waters® UPLC aligned with a Waters® Xevo–TQ-SMS mass spectrometer (Waters Ltd., UK). MassLynx® 4.1 software was used for data acquisition and analysis. Liquid chromatography separation was performed on an Acquity® UPLC CSH C18 (50 × 2.1 mm, 1.7 μm particle size) column at 50 °C. The mobile phase consisted of acetonitrile spiked with 0.1% formic acid and 10 mM ammonium formate solution. The mobile phase was flowed at a rate of 0.6 mL/min using a time and solvent gradient composition. The initial composition (100% aqueous) was held for 1 min, following which the composition was changed linearly to 20% organic phase over the next 0.25 min. The composition was then brought to 100% organic over 1.25 min, held for 0.5 min and finally shifted to the initial composition and held constant until the end of the run for column equilibration. The total run time was 4 min and sample injection volume was 2.5 μL. The mass spectrometer was operated in the multiple reaction monitoring (MRM) mode. Sample introduction and ionization was by electrospray ionization (ESI) in the positive ion mode. DTG, RPV and CAB were analysed at mass-to-charge ratios of 420.16, 406.08 and 367.13 respectively. Stock solutions of DTG, RPV, CAB, and an internal standard maraviroc were prepared in methanol at a concentration of 500 μg/mL. A ten-point calibration curve was prepared ranging from 2.5–5000 ng/mL. Quality control samples were prepared in a similar procedure using an independent stock solution at three concentrations (5, 50 and 500 ng/mL).

Each sample was analysed three times. Drug concentrations were considered non-zero only if the drug concentrations were above the lower limit of detection in each of the three analyses. All formulations were tested in three pigs based on our previous studies in the large animal model[42].

**Acquisition and analyses of photographs of the dosage forms.** Photographs of gastric resident dosage forms prior to dosing and after recovery from the animals were acquired using a VHX-5000 Digital Microscope (Keyence Corp., U.S.A.) equipped with VH-Z20R/Z20W/Z20T lenses. Photographs were obtained by automatically stitching parts of the image at a 10 μm step size and were analysed and processed using an integrated Keyence image analysis software.

**Data availability.** The data supporting the findings of this study are available within the article and its Supplementary Information files and from the corresponding authors on reasonable request.

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

## Acknowledgements

We want to thank S. Kern, D. Hartman, S. Hershenson and W. H. Gates from the Bill and Melinda Gates Foundation for helpful discussions around the application and development approach of the extended release gastric resident system with respect to HIV. We thank J. Haupt and M. Jamiel for help with the in vivo porcine work. We thank P. Lozano, C. Fucetola and D. Freeman for assistance with microscopy. We thank C. Settens for X-ray diffraction analyses of the drugs. We are grateful to all members of the Langer and Traverso Laboratories for helpful discussions around gastric resident systems and in particular to G. Gaiha.
Funding: This work was funded in part by the Bill and Melinda Gates Foundation Grants No. OPP1139937, OPP1139921 and OPP1148627, NIH Grants No. EB-000244, R01AI131416 and DP5OD019851. G. Traverso was supported in part by the Division of Gastroenterology, Brigham and Women's Hospital. C. Selinger, A. Bershteyn and P. A. Eckhoff are supported by Bill and Melinda Gates through the Global Good Fund.

## Author contributions

A.R.K, O.A, L.W., B.N., R.L., and G.T. conceived and designed the research. O.A., D.M., and T.B. performed the mechanical characterization. L.B., C.C. and A.H. performed the in vivo pig experiments. Synthesis and characterization of polymer and drug formulations were performed by A.R.K., O.A., Y.-A.L.L. and J.R. Synthesis, in vitro and in vivo characterization of dosage forms were performed by O.A., A.R.K, T.B, H.M., F.J. and S.J.W. Linker synthesis was performed by A.M.B. and T.G. Epidemiological modelling was performed by C.S., A.B. and P.A.E. A.H., M.C., S.S.M., and M.A.N. performed viral dynamics modelling. A.R.K., O.A., R.L., and G.T. analysed the data. A.R.K., G.T., and R. L. wrote the manuscript. D.M. and T.B. contributed equally to this work with their efforts focused on platform prototyping and mechanical characterization.
