## [Peer Review File · Nature Communications]

Reviewers' comments:

Reviewer #1 (Remarks to the Author):

Comments and Recommendations

1. The authors' should explain their choice of antiretroviral agents (ARVs) selected for this study. No clinical study has been conducted or is underway (that this reviewer can find) with the combination evaluated (DTG + CAB + RPV + TAF). Additionally, no pharmacologic basis for dual integrase inhibitor therapy (e.g. DTG + CAB) has been advanced (to this reviewer's knowledge). Thus the question of why this combination, as the current state of HIV therapeutics would indicate it isn't clinically relevant?
2. Salt forms of the ARVs studied were synthesized. What information do the investigators' have that their synthesized compound has similar in vivo pharmacokinetic (PK) behavior (e.g., bioavailability, absorption) versus the commercially available drug?
3. The PK results for DTG, CAB and RPV from the immediate release do not mimic those concentrations achieved in humans taking the usual oral dose. For example, with the immediate release formulation of DTG, a maximum concentration (C_{max}) of approximately 500 ng/mL was achieved. In humans, the usual C_{max} is 3700 ng/mL. Similarly, for CAB and RPV, the immediate release formulations produced approximate average C_{max} values of 1400 ng/mL and 25 ng/mL, respectively, were achieved, compared with C_{max} values of 2800 ng/mL and 200 ng/mL, respectively, in humans receiving the usual dose. This raises questions potentially about synthesis of the oral formulations (#2), a dose not optimally allometrically scaled, or quite different PK in pigs vs. humans that would raise questions about the ability to extrapolate the PK data to humans. Additionally comment on the considerable inter-animal variability in CAB concentrations (500 to almost 3000 ng/mL) should be made.
4. For oral PK, a solution of rilpivirine was formulated and placed in a gelatin capsule. This isn't clear – would not the capsule begin to dissolve immediately, even before it could be administered to the pig?
5. Atropine can inhibit gastric acid production and gastrointestinal motility. Are there any effects of the atropine dose used for anesthesia on the oral PK of the drugs studied?
6. I am concerned about the value of the simulation of weekly formulation for pre-exposure prophylaxis (PrEP) to this paper. None of the drugs used in the weekly formulation have established (i.e., FDA approved) efficacy and safety for PrEP. Importantly, this includes TAF, while even though it is converted to the same active moiety as tenofovir disoproxil fumarate (TDF), the mucosal tissue PK of TAF is not equivalent in mucosal tissues in humans to TDF (see Cottrell M, J Antimicrob Chemother, March 2017, <https://www.ncbi.nlm.nih.gov/pubmed/28369415> . Thus, this really becomes just a simulation exercise, is not informed based on established PrEP efficacy for the drugs studied, and does not use any of the PK data generated in the present study. Given that, the conclusions that "hundreds of thousands of HIV cases could be avoided" is not evidence based from the results of this paper with the weekly formation. I believe the PrEP simulation component of the paper should be removed.
7. Supplementary Figure 4 gives the usual ARV doses, which are readily available. It can be deleted.
8. On Figure 1B, the authors need to include units on the listed dimensions.

9. The description of "ultra" long acting is really in the eye of the beholder. Weekly administration is clearly not ultra long if contrasted with, for example, hormonal implants. This description should be deleted (Abstract, Introduction and elsewhere).

10. In the discussion, the authors' state "high and relatively consistent" plasma drug concentrations were achieved. Again, "high" is in the eye of the beholder. What the authors' do need to acknowledge, as noted in comment #3, is that the concentrations achieved do not mimic those achieved in humans that for DTG and RPV have been shown safe and effective (FDA approved) for treatment of HIV infection.

11. The authors' state their work is the development of the first orally-available sustained release ARV. This might be true, however, I'm uncertain enough to raise a concern about this priority claim. Several other groups are working on novel oral formulations of ARVs. See for example, the work by Owen with an oral nanoformulation in humans that allowed a 50% dose reduction yet achieved the same systemic exposure:

<http://www.croiconference.org/sessions/human-confirmation-oral-dose-reduction-potential-nanoparticle-arv-formulations> I feel some acknowledgement should be made of other work with novel oral formulations of ARVs.

Reviewer #2 (Remarks to the Author):

This paper by Traverso and colleagues represents another seminal contribution to the design of long-lasting gastro-retentive oral drug delivery systems. It is distinct from their paper in Sci Transl Med in 2016 in several ways, including an innovative approach that allows use of the best polymers to ensure structural integrity (thus allowing gastro-retention) in addition to the best polymers for drug release (to achieve the desired PK profiles), thus decoupling these two critical features. The system has six arms that can each be loaded with different drugs and/or polymer systems. It is not critical whether the polymers used are the optimal ones at this stage, especially since the authors clearly show impressive proof-of-concept of the flexibility of the system in this regard. The studies were carefully done and significant pharmacokinetics of multiple drugs in the pig model are demonstrated. It is clear that this work could have implications well beyond HIV maintenance therapy and HIV PREP.

A few minor questions:

- 1) how much slower is GI transit in pigs v. humans? A more thorough discussion would be helpful.
- 2) is there any concern about certain types of food that may be more likely to cause device failure (the pigs were fed "pellets" and fruits/veggies... these seem like they would easily be reduced in the stomach and thus easily pass by the delivery system without causing damage).
- 3) how easy will it be to include a pH sensitive linker in the system intended to be moved forward, as opposed to the proof-of-concept system shown? Will retention in the GI tract for six days cause any problems in humans? Will degradation of the pH-sensitive linker still occur to some degree in the lower pH of the stomach?
- 4) what would be the expected impact on therapy or protection in the (hopefully rare) case

that a system passes into the GI tract prematurely?

5) what size pill will be required to load these three drugs into one system for weekly administration?

Reviewer #3 (Remarks to the Author):

In their manuscript "Development of an oral once-weekly drug delivery system for HIV antiretroviral therapy", Kirtane, Abouzid et al. designed a new delivery system of antiretrovirals against HIV infection, tested its physical properties, analysed the pharmacokinetics in a pig model system and modelled the potential impact of such a system on a within-host level and on an epidemic scale. The goal of such a delivery system is to increase the adherence to the drug regiment as one would expect that individuals are more likely to take a pill once every week instead of every day. The idea of a polymer delivery system that releases drugs slowly seems very promising to me. The manuscript is well written and the study seems timely and important. In addition, I very much like the combination of establishing the drug delivery system experimentally and analysing the potential impact employing a theoretic framework. To my knowledge this approach is new and could serve as a potential improvement of existing drug regiments. Unfortunately, I cannot comment on the experiments in greater detail as I am a theoretic biologist, hoping that one of the other reviewers is more skilled in judging this part of the manuscript. My review focusses on the modelling part of the paper. Here, I am missing many details that were necessary for carefully examining the validity of the modelling approach. This is why I recommend major revisions. My criticism in detail:

Major points

Within-host level

1. The employed mathematical framework to estimate treatment failure, first introduced by Rosenbloom et al Nat. Med. 2012., is based on the determination of the mutant selection windows of specific resistance mutations against specific drugs. As it is essential to this framework to exactly know which specific escape variants and which specific ancestral strains are compared, it is important to report these strains also in the current manuscript. Each infected individual might have a different ancestral strain from which an escape variant can form and thus, different drug adherence schemes would result in different escape predictions for each individual. Please list the exact escape variants and discuss these variants in the light of different patients.
2. Along the lines of point 1: For being able to use this specific framework, one needs to know the IC50 values and the Hill-slope parameter of the in vitro measured inhibition curves of the ancestor/escape variants. These values need to be listed in the manuscript. Furthermore it is important to know the fitness differences between the escape variants and the ancestral HIV strain. How was this difference measured or estimated? Please add a table with all IC50 values, slope values and fitness differences.
3. Supplement line 323: The authors assume that the half-life of the drug is seven times higher when employing the new drug delivery system in comparison to a daily pill. How is

this assumption justified? Even if the pharmacokinetics were only measured in pigs would it not be more reasonable to calculate the half-lives in this system and basing this important parameter for estimating drug failure than just assuming a somehow random number?

4. In the light of reproducibility I have to say that I find the description of the within-host level very rudimentary and would advise the authors to extend this section, especially because the model description is already in the supplement. A mathematical model description requires in my view also quoting the used mathematical formula.

Epidemic scale

5. The model for estimating the impact of the new dosing scheme used as PrEP on the spread of HIV are also not described in sufficient detail. What does the employed EMOD do exactly, how is it parameterised? What is the distinction between the assumptions 90-90-90 and ART? What do these regimens have to do with PrEP efficacy?

6. Supplement: line 374: I am not sure where the authors obtain the estimates on PrEP efficacy levels of 50%. Does that mean, when a person is on PrEP, the probability of becoming infected is 0.5 times the probability to become infected when not on PrEP? On the cdc webpage <https://www.cdc.gov/hiv/basics/prep.html>, the efficacy is estimated with 70-90%.

Minor points

7. Please add a table to the supplement with all abbreviations used if the journal allows that.

8. Supplement: line 303 there is one “\” where it does not belong

9. Manuscript: line 329 should be “dosage form” instead of “dosage dorm”

10. Figure 1D: Why are the plasma concentrations this high and constant over time? If you just look at the sum of the individual concentrations, one would not see this constant level. Are the green dots measured values?

11. Figure 2ABC and Figure 3B: Do not use histograms when actually representing three independent replicates. Instead it is better practise to show the actual measurements. This is much more meaningful than a mean with standard variation.

12. Figure 3A, CDE; Figure 4 and Supplement Figure 6: Also in these figures it would be better to show the individual measurements (again n=3) instead of the mean. The reader would obtain a much clearer view on the actual processes and especially in Figure 4 and SuppFig 6 could compare the outcomes in different animals.

Point-by-Point Response

Reference: NCOMMS1708538

We are extremely grateful to the Reviewers for their time, and appreciate their comments and recommendations for improving the quality of our manuscript. Below we address the Reviewers' comments and based on their suggestions, we have made several changes to the manuscript.

Comments and Recommendations

Reviewer 1

- 1. The authors' should explain their choice of antiretroviral agents (ARVs) selected for this study. No clinical study has been conducted or is underway (that this reviewer can find) with the combination evaluated (DTG + CAB + RPV + TAF). Additionally, no pharmacologic basis for dual integrase inhibitor therapy (e.g. DTG + CAB) has been advanced (to this reviewer's knowledge). Thus the question of why this combination, as the current state of HIV therapeutics would indicate it isn't clinically relevant?**

We thank the Reviewer for bringing up this very important point. The choice of drugs was motivated, at least in part, by the mounting evidence supporting their efficacy alone or in combination (integrase inhibitors + non-nucleoside reverse transcriptase inhibitors) for maintenance therapy as well as the potency of the drugs as reflected in Supplementary Figure 4 given the volume limitations of a capsule.

Efficacy of cabotegravir-rilpivirine combination as maintenance therapy in patients infected with HIV-1 was reported recently¹. In this Phase 2b clinical trial, patients with suppressed viral load were treated with either cabotegravir plus rilpivirine or efavirenz plus dual nucleoside reverse transcriptase inhibitors. The study concluded that the cabotegravir-rilpivirine combination was safe and was effective at maintaining suppressed viral load. Other clinical trials are underway to evaluate safety and tolerability of a combination of long acting formulations of cabotegravir and rilpivirine in healthy individuals (clinicaltrials.gov identifier: NCT01593046) and their efficacy (clinicaltrials.gov identifier: NCT02938520) in patients that have achieved viral suppression with an integrase inhibitor-based single tablet regimen.

The efficacy of dolutegravir (as monotherapy or dual therapy with rilpivirine) in maintaining suppressed viral loads in treatment experienced patients has been shown in a few clinical studies as well²⁻⁶. In these studies, after a two-year follow up, >95% patients showed plasma viremia levels of <50 copies/mL. Additional clinical trials are underway studying the effect of switching to a dolutegravir + rilpivirine in patients with suppressed viremia when treated with 2 nucleoside reverse transcriptase inhibitors and another drug (clinicaltrials.gov identifier: NCT02422797).

In addition to these clinical trials, studies in macaques have shown that treatment with the long acting formulation of cabotegravir protects the animals from simian immunodeficiency virus infection^{7,8}. Safety of the cabotegravir formulation⁹ and a long acting formulation of rilpivirine¹⁰ in humans has also been established.

We would not suggest combining DTG and CAB as these drugs are analogs.

Drug potency is essential for maximizing the amount of active pharmaceutical ingredient that can be packaged in a dosage form. Given the volume of the largest commonly used capsule is approximately 1.3 cm³ the theoretical maximum load for a density of 1g/ml is approximately 1g. Supplementary Figure 4 presents the range of antiretrovirals with respect to their dose highlighting the paucity for potent antiretrovirals. We have emphasized this point in the *Methods* section.

In light of this evidence, we believe that our choice of drugs is relevant, and will be of considerable interest to the field. We have now included this in the *Discussion* section of the manuscript (Main document, Page 14, first paragraph on the page and Main document, Page 17, middle of the page).

2. Salt forms of the ARVs studied were synthesized. What information do the investigators' have that their synthesized compound has similar in vivo pharmacokinetic (PK) behavior (e.g., bioavailability, absorption) versus the commercially available drug?

The commercially available dosage forms of these drugs (viz. Tivicay[®] and Edurant[®]) contain the same salt forms as used in our study. In fact, the presence of the drug in their salt forms in the marketed formulations was our motivation to generate salt forms of these drugs. We have now added this rationale to the *Methods* section of the manuscript (Supplemental information I, Page 4, top of the page).

3. The PK results for DTG, CAB and RPV from the immediate release do not mimic those concentrations achieved in humans taking the usual oral dose. For example, with the immediate release formulation of DTG, a maximum concentration (C_{max}) of approximately 500 ng/mL was achieved. In humans, the usual C_{max} is 3700 ng/mL. Similarly, for CAB and RPV, the immediate release formulations produced approximate average C_{max} values of 1400 ng/mL and 25 ng/mL, respectively, were achieved, compared with C_{max} values of 2800 ng/mL and 200 ng/mL, respectively, in humans receiving the usual dose. This raises questions potentially about synthesis of the oral formulations (#2), a dose not optimally allometrically scaled, or quite different PK in pigs vs. humans that

would raise questions about the ability to extrapolate the PK data to humans. Additionally comment on the considerable inter-animal variability in CAB concentrations (500 to almost 3000 ng/mL) should be made.

We thank the reviewer for highlighting this point. We agree with the reviewer that there are disparities in the concentrations observed in our studies and those observed in clinical studies and we acknowledge this in the *Discussion* section as one of the limitations of our study.

We believe the differences in concentration could be attributed to the animal model used in our study. Previous work has highlighted differences in pharmacokinetics between pigs and humans¹¹. For example, when dosed orally in pigs the concentrations of cyclosporine A and prednisolone achieved are about one tenth of those observed in humans¹². This has been attributed to a higher intrinsic clearance by liver enzymes and a higher volume of distribution in pigs. Despite these differences, for the current study, the swine model was particularly well suited. The pig stomach resembles the human stomach in size and anatomical features. Moreover, the average weight of the pigs used in our studies was similar to that of adult humans. As a significant portion of our study was focused on developing a gastric retentive system, we chose a swine model. The inter-animal variability is likely because all dosage were prepared by hand and mechanized manufacturing will significantly aid in reducing these variabilities. Future efforts in other species including dogs and non-human primates, and with dosage forms that are machine manufactured will be essential in translating this work to human application. We note this in the *Discussion* section of the manuscript (Main document, Page 16, middle of the page). It should be noted though, as Reviewer 2 points out, because the formulation loaded on this system can be changed independent of the platform dosage form, if future studies reveal that concentrations are inappropriate, alterations in the formulation could be made with relative ease.

- 4. For oral PK, a solution of rilpivirine was formulated and placed in a gelatin capsule. This isn't clear – would not the capsule begin to dissolve immediately, even before it could be administered to the pig?**

We have clarified this point in the *Methods* section (Supplemental information I, Page 11, middle of the page). We did in fact observe capsule dissolution upon placing the solution in it, however it takes a few minutes for the capsule to dissolve. Hence, we loaded the drug solution in the capsule immediately prior to dosing it to the animal. As the time between capsule filling and dosing was very short (few seconds), we were able to dose accurately.

- 5. Atropine can inhibit gastric acid production and gastrointestinal motility. Are there any effects of the atropine dose used for anesthesia on the oral PK of the drugs studied?**

We thank the reviewer for highlighting this point. We have checked several databases (www.rxlist.com, www.medscape.com, www.drugs.com and www.micromedexsolutions.com) and there have been no interactions reported between dolutegravir and atropine or rilpivirine and atropine. There is currently less information available about cabotegravir as it is not yet FDA approved.

The oral pharmacokinetics of the drugs especially those in the gastric retentive dosage form was evaluated for a period of days. The terminal half-life of atropine is approximately 2 hours¹³, hence we believe atropine would have rather minimal effect on the overall findings of the pharmacokinetic study. We have included the known half-life of atropine in pigs in the *Methods* section (Supplemental information I, Page 12, top of the page).

- 6. I am concerned about the value of the simulation of weekly formulation for pre-exposure prophylaxis (PrEP) to this paper. None of the drugs used in the weekly formulation have established (i.e., FDA approved) efficacy and safety for PrEP. Importantly, this includes TAF, while even though it is converted to the same active moiety as tenofovir disoproxil fumarate (TDF), the mucosal tissue PK of TAF is not equivalent in mucosal tissues in humans to TDF (see Cottrell M, J Antimicrob Chemother, March 2017, <https://www.ncbi.nlm.nih.gov/pubmed/28369415> . Thus, this really becomes just a simulation exercise, is not informed based on established PrEP efficacy for the drugs studied, and does not use any of the PK data generated in the present study. Given that, the conclusions that “hundreds of thousands of HIV cases could be avoided” is not evidence based from the results of this paper with the weekly formation. I believe the PrEP simulation component of the paper should be removed.**

We thank the reviewer for highlighting this point that we have now addressed in the manuscript. Our manuscript presents technologies enabling the delivery of antiretrovirals that could be dosed once a week. Given prior work suggesting the potential benefits with respect to medication adherence^{14,15} we felt it important to understand the relationship between potential adherence improvement with long-acting therapy and how this might impact patients infected with HIV, particularly given the high levels of non-adherence in these patient populations.

For our modeling, we used TDF clinical data as that is one of the only HIV drugs that has been extensively studied in the PrEP setting. Consequently, we have robust data to understand the complex relationship between patient adherence and efficacy of PrEP. An implicit assumption of our modeling is that any new drug that would be approved for PrEP use in the general population would be at least as effective as current TDF-based prevention. However, to account for potential differences, we consider a wide range of efficacy values in our sensitivity analysis. Once the pharmacology and efficacy of the drugs we used in our studies is established, such a framework can be applied to those drugs as well. In our current models we estimate:

- the expected increase in adherence to PrEP that would accompany long-acting therapy, and in turn, the expected reduction in infection rate of PrEP-treated individuals (assuming TDF-like adherence-efficacy relationships)
- the expected decrease in new HIV infections over a 20-year period in a South Africa-like setting, given this estimated increase in PrEP efficacy

We have expanded the points above and limitations of the modeling in the *Discussion* section (Main text, Page 15, bottom of the page).

Future work, which is beyond the scope of this paper, will combine new drug-specific data and within-host models that can estimate how changes in pharmacokinetics and pharmacodynamics influence PrEP efficacy, with the EMOD population model, to obtain more scenario-specific estimates for the impact of PrEP.

Although the drugs evaluated in this study – RPV, DTG, and CAB – have not yet completed human efficacy trials for use as pre-exposure prophylaxis, many in the field have discussed their potential promise for this use and trials are planned. For example, CAB, in its long-acting injectable form, has been evaluated in the macaque-SIV model of HIV infection and shown to be able to prevent both rectal and vaginal transmission^{7,8}. Two major clinical trials are underway to evaluate its efficacy as PrEP, compared to TDF/FTC, in both women and men (clinicaltrials.gov indicators NCT03164564 and NCT02720094). RPV is also being evaluated for prophylaxis¹⁶. When delivered either encapsulated nanoparticles forming a vaginal gel, or, as crystalline nanoparticles in an intramuscular injection, RPV can protect humanized mouse from high dose vaginal transmission¹⁷. Phase II clinical trials to evaluate the safety and acceptability of long-acting injectable RPV for eventual use as PrEP are currently underway (HPTN 076, clinicaltrials.gov indicator NCT02165202). Existing studies have shown that RPV reaches high levels in vaginal and rectal tissue¹⁸. DTG has not yet been explicitly evaluated for PrEP, but has been shown to be effective as maintenance monotherapy^{2,3} and for post-exposure prophylaxis¹⁹, has been shown to have suppressive levels in the female genital tract and male rectal tissue^{20,21}, and has been incorporated into nanoparticles to form a vaginal gel for preventing transmission²². Based on this extensive body of research-in-progress, we feel it is important to consider the potential public health implications of the improved adherence potential of long-acting ART.

7. Supplementary Figure 4 gives the usual ARV doses, which are readily available. It can be deleted.

We thank the reviewer for bringing this to our attention and have clarified in the *Methods* section the rationale for inclusion (Supplemental information I, Page 4, top of the page). As noted under the first point from this reviewer we included Supplementary Figure 4 in the manuscript as it

explained, in part, our motivation behind the choice of drugs for this paper. Moreover, other efforts are also underway to develop injectable long acting antiretroviral systems. These systems, similar to ours, have been developed for only those drugs that have daily doses of <50 mg/day. Hence, we believe inclusion of this figure highlights this limitation of long-acting systems and could motivate the development of more potent ARVs to allow broader applications of such systems.

8. On Figure 1B, the authors need to include units on the listed dimensions.

We thank the Reviewer for bringing this to our attention. We have now included the dimensions in Figure 1B.

9. The description of “ultra” long acting is really in the eye of the beholder. Weekly administration is clearly not ultra long if contrasted with, for example, hormonal implants. This description should be deleted (Abstract, Introduction and elsewhere).

We have removed “ultra” from the text.

10. In the discussion, the authors’ state “high and relatively consistent” plasma drug concentrations were achieved. Again, “high” is in the eye of the beholder. What the authors’ do need to acknowledge, as noted in comment #3, is that the concentrations achieved do not mimic those achieved in humans that for DTG and RPV have been shown safe and effective (FDA approved) for treatment of HIV infection.

We agree with the reviewer. In the revised manuscript, we now do not use the word “high” in this context. Moreover, we highlight the fact that there is a difference in the concentrations we observe in our study and those observed in clinical trials (Main document, Page 16, middle of the page).

11. The authors’ state their work is the development of the first orally-available sustained release ARV. This might be true, however, I’m uncertain enough to raise a concern about this priority claim. Several other groups are working on novel oral formulations of ARVs. See for example, the work by Owen with an oral nanoformulation in humans that allowed a 50% dose reduction yet achieved the same systemic exposure: <http://www.croiconference.org/sessions/human-confirmation-oral-dose-reduction-potential-nanoparticle-arv-formulations> I feel some acknowledgement should be made of other work with novel oral formulations of ARVs.

We thank the reviewer for highlighting this very important contribution. We have now added a paragraph describing this work in the *Discussion* section (Main text, Page 13, bottom of the page).

Reviewer #2 (Remarks to the Author):

This paper by Traverso and colleagues represents another seminal contribution to the design of long-lasting gastro-retentive oral drug delivery systems. It is distinct from their paper in *Sci Transl Med* in 2016 in several ways, including an innovative approach that allows use of the best polymers to ensure structural integrity (thus allowing gastro-retention) in addition to the best polymers for drug release (to achieve the desired PK profiles), thus decoupling these two critical features. The system has six arms that can each be loaded with different drugs and/or polymer systems. It is not critical whether the polymers used are the optimal ones at this stage, especially since the authors clearly show impressive proof-of-concept of the flexibility of the system in this regard. The studies were carefully done and significant pharmacokinetics of multiple drugs in the pig model are demonstrated. It is clear that this work could have implications well beyond HIV maintenance therapy and HIV PREP.

A few minor questions:

- 1. How much slower is GI transit in pigs v. humans? A more thorough discussion would be helpful.**

This is an important point and one that we strongly consider in all of our studies. Including these considerations in the manuscript will be a valuable addition, which we have included in the *Discussion* section (Main text, Page 16, top half of the page).

- 2. Is there any concern about certain types of food that may be more likely to cause device failure (the pigs were fed "pellets" and fruits/veggies... these seem like they would easily be reduced in the stomach and thus easily pass by the delivery system without causing damage).**

We thank the reviewer for highlighting this point. We agree that understanding the relationship between food and novel dosage forms is a very important one. The pigs we worked with are fed a broad diet. We have clarified in the *Methods* section (Supplemental information I, page 11, first paragraph on that page) the breadth of foods they receive during study periods and also acknowledge in the *Discussion* section that evaluation of food effects will be an important aspect in successful human translation (Main text, Page 16, top half paragraph of that page).

3. How easy will it be to include a pH sensitive linker in the system intended to be moved forward, as opposed to the proof-of-concept system shown? Will retention in the GI tract for six days cause any problems in humans? Will degradation of the pH-sensitive linker still occur to some degree in the lower pH of the stomach?

To include pH sensitive linkers into the dosage form, we solvent-weld an alkaline soluble polymer film onto the arms of the dosage form. This technique can potentially be applied to the current dosage form as long as a common solvent for the major component of the film and the arm is available. Hence, we believe that with some optimization such linkers can be included into the current dosage form as well. With respect to scale up we anticipate manufacturing dosage forms using injection molding and have demonstrated the capacity to extrude and injection mold the various materials used. We have now included this description in the *Methods* section (Supplemental information I, page 9, bottom half).

In our experiments, we introduced an entire intact dosage form directly into the intestine to simulate the worst case scenario for these systems. Even in this extreme case, we did not observe any adversity to the animals. We believe this is because the dosage form weakens rapidly and considerably in the intestine, and starts dissociating into its constitutive arms. Retention of arms in the small intestine is not expected to produce any adverse effects as it is a two-dimensional object with one of its dimensions smaller than the ileocecal sphincter (the narrowest portion of the intestine).

We have previously evaluated the stability of the linker in simulated gastric fluid and simulated intestinal fluid *in vitro*²³. Our tests reveal that after 7 days, the adhesion force of the linkers is reduced by only ~10% in simulated gastric fluid. In contrast, in simulated intestinal fluid, the adhesion force of the linkers is reduced by 100% over the same time span. This suggests that the linkers are resistant to degradation in acidic environments over the time scales of interest. However, further *in vivo* evaluation will be required as the *in vitro* test does not completely capture all nuances of an *in vivo* environment.

4. What would be the expected impact on therapy or protection in the (hopefully rare) case that a system passes into the GI tract prematurely?

The impact of premature release of either the entire dosage form or an individual arm into the small intestine depends on multiple factors. The small intestine is the site of absorption of oral drugs, and so drug absorption would continue. The total drug exposure would depend on any differences in drug release from the dosage form in the intestine vs the stomach, which would likely change depending on the drug-polymer formulation. Drug exposure would also depend on the transit time of the prematurely-released device through the intestine, which will result in less time for absorption than if this device piece had remained in the stomach for the full dose

interval. These effects, together with the timing of the premature device passage, would determine any potential effects on the pharmacokinetic profile, which would in turn influence the effect on suppression of viral replication as well as on potential drug toxicity. We hope to address these issues in much more detail once reproducible, industrial-grade versions of the dosage form have been created and tested in animal model under a larger variety of circumstances.

In the current manuscript, we specify that better understanding of the above-mentioned factors is important to guide formulation design and treatment strategy (Main document, Page 17, top of the page)

5. What size pill will be required to load these three drugs into one system for weekly administration?

The use of all three drugs together in one patient is unlikely. If used for PrEP, only one drug is sufficient, and for such an application two-three capsules are required per week, and these can be administered simultaneously. For treatment with dual therapy, five-six capsules will be required per week which can be administered simultaneously. We have included this discussion in the manuscript now (Main document, Page 14, bottom half of the page). These calculations are based on hand-based fabrication of the dosage form. Manufacturing the polymer-drug formulations by extrusion may lead to greater compaction and hence may require fewer number of capsules.

Reviewer #3 (Remarks to the Author):

In their manuscript “Development of an oral once-weekly drug delivery system for HIV antiretroviral therapy”, Kirtane, Abouzid et al. designed a new delivery system of antiretrovirals against HIV infection, tested its physical properties, analysed the pharmacokinetics in a pig model system and modelled the potential impact of such a system on a within-host level and on an epidemic scale. The goal of such a delivery system is to increase the adherence to the drug regimen as one would expect that individuals are more likely to take a pill once every week instead of every day. The idea of a polymer delivery system that releases drugs slowly seems very promising to me. The manuscript is well written and the study seems timely and important. In addition, I very much like the combination of establishing the drug delivery system experimentally and analysing the potential impact employing a theoretic framework. To my knowledge this approach is new and could serve as a potential improvement of existing drug regimens. Unfortunately, I cannot comment on the experiments in greater detail as I am a theoretic biologist, hoping that one of the other reviewers is more skilled in judging this part of the manuscript. My

review focusses on the modelling part of the paper. Here, I am missing many details that were necessary for carefully examining the validity of the modelling approach. This is why I recommend major revisions. My criticism in detail:

Major points

Within-host level

1. **The employed mathematical framework to estimate treatment failure, first introduced by Rosenbloom et al Nat. Med. 2012., is based on the determination of the mutant selection windows of specific resistance mutations against specific drugs. As it is essential to this framework to exactly know which specific escape variants and which specific ancestral strains are compared, it is important to report these strains also in the current manuscript. Each infected individual might have a different ancestral strain from which an escape variant can form and thus, different drug adherence schemes would result in different escape predictions for each individual. Please list the exact escape variants and discuss these variants in the light of different patients.**

Thank you for pointing out this omission. We have now included a detailed supplementary methods section that describes all details of our model, including all resistant strains that were considered for each drug, the parameters associated with them, and their source in the literature. Throughout the paper, we consider that all patients start treatment with fully wild-type virus, with other mutations existing only at a level given by mutation-selection balance. So, for example, if a possible resistance mutation is K65R, then we assume all patients start with a viral sequence that (mostly) codes for lysine (K), and may later gain mutations that convert this sequence to arginine (R) (or, selection may act on pre-existing mutations with this property). The specific base pairs underlying these codons is chosen based on the base composition of the HIV consensus genome. Further details are provided in the Supplementary Methods. While in reality, participants may have more heterogeneous viral sequences in which some resistance mutations may have been transmitted or in which some neutral intermediates to resistance mutations may exist at high frequencies, considering all these complexities is beyond the scope of the current modeling contribution to this project. We are in the process of preparing a separate manuscript in which many more realistic aspects of treatment will be included in the model, such as inter-patient variation in parameters, more detailed pharmacokinetic models, combination treatment, and pre-exposure prophylaxis.

2. **Along the lines of point 1: For being able to use this specific framework, one needs to know the IC50 values and the Hill-slope parameter of the in vitro measured inhibition curves of the ancestor/escape variants. These values need to be listed in the manuscript. Furthermore it is important to know the fitness differences between the escape variants and the ancestral HIV strain. How was this difference measured or estimated? Please add a table with all IC50 values, slope values and fitness differences.**

We apologize for not including all these details with the initial submission. The new *Supplementary Methods* now details all of these parameters as well as the methods used to estimate them, with links the literature source for each value. In summary, where possible, all values for IC50 and slope of both wild-type and mutant strains was taken from experiments conducted in the laboratory of Dr. Robert Siliciano (referenced in the *Supplementary Methods*). They use a single-round infectivity assay conducted in primary CD4 T cells from donors, in donor plasma, with a reporter virus encoding GFP, into which they can introduce point mutations in a constant genetic background. However, some important resistance pathways have not been considered by this lab, and so we also included data from other groups, including those of Dr. Mark Wainberg, Monogram Biosciences, or Shionogi Biosciences, which use related methods.

- 3. Supplement line 323: The authors assume that the half-life of the drug is seven times higher when employing the new drug delivery system in comparison to a daily pill. How is this assumption justified? Even if the pharmacokinetics were only measured in pigs would it not be more reasonable to calculate the half-lives in this system and basing this important parameter for estimating drug failure than just assuming a somehow random number?**

Thank you for pointing out this potential source of confusion. We now include a more detailed justification for this choice in the *Supplementary Methods*:

“For long-acting formulations, we consider a hypothetical scenario. We think that the pharmacokinetic profiles observed in the swine model in this preliminary study are likely to be different than those observed in eventual human testing, both because drug absorption and elimination is species-specific, and, because the current dosage forms are hand-made in the laboratory but will undergo extensive improvements and industrial-grade manufacturing before administration to humans. Therefore, we did not focus on characterizing the swine pharmacokinetics in detail and exactly reproducing these in our model. Instead, we assumed a relatively simple profile for a hypothetical clinical long-acting formulation. Since the form will be given once a week, we assumed that the half-life was seven-fold longer than the existing formulation. As a baseline case, we assumed the peak concentration C_{\max} was the same, but also explored lower values. Otherwise the parameters for long-acting antiretrovirals were assumed to be the same as for the daily formulation.”

Overall, we think a choice of seven-fold increase in half-life for a seven-fold increase in dose interval is reasonable, because it’s likely that eventual clinical approval of such a device would require that the half-life be extended by an amount commensurate with the proposed increase in time between administrations. We are in the process of preparing a separate manuscript focused exclusively on the modeling which will explore a larger range of scenarios. We are also designing more detailed pharmacokinetic studies in the swine model to then be able to more

accurately characterize the half-life in this system, and, we will be continuing to incorporate modeling into the evaluation of the device as it proceeds through development to eventual human testing.

- 4. In the light of reproducibility I have to say that I find the description of the within-host level very rudimentary and would advise the authors to extend this section, especially because the model description is already in the supplement. A mathematical model description requires in my view also quoting the used mathematical formula.**

We apologize for the absence of these important details in the original submission. We have now included a very detailed *Supplementary Methods* that includes all formulas, algorithms, and parameters needed to reproduce the model.

Epidemic scale

- 5. The model for estimating the impact of the new dosing scheme used as PrEP on the spread of HIV are also not described in sufficient detail. What does the employed EMOD do exactly, how is it parameterised? What is the distinction between the assumptions 90-90-90 and ART? What do these regimens have to do with PrEP efficacy?**

We thank the reviewer for pointing this out. In the revised version of the manuscript, we have now included the following description in the *Methods* section (*Supplementary Information I*, Page 15, bottom half):

“EMOD-HIV is an individual-based model that simulates transmission of HIV using an explicitly defined network of heterosexual relationships that are formed and dissolved according to age- and risk-dependent preference patterns²⁴. The synthetic population was initiated in 1960, and population recruitment and mortality was assumed to be proportional following age- and gender-stratified fertility and mortality tables and projections from the 2012 UN World Population Prospects. The model was calibrated to match retrospective estimates of age-stratified, national-level prevalence, incidence, and ART coverage from four nationally representative HIV surveys in South Africa. The age patterns of sexual mixing were configured to match those observed in the rural, HIV-hyperendemic province of KwaZulu-Natal, South Africa²⁵. Recently, a validation study showed that self-reported partner ages in this setting are relatively accurate, with 72% of self-reported estimates falling within two years of the partner’s actual date of birth²⁶. Further, the transmission patterns observed in EMOD are consistent with those revealed in a recent phylogenetic analysis of the age/gender patterns of HIV transmission in this setting²⁷.

Transmission rates within relationships depend on HIV disease stage, male circumcision, condom usage, co-infections, and antiretroviral therapy^{28,29} until achieving a reduction in transmission of 92%-an estimate based on observational data of serodiscordant couples in which outside partnerships could have contributed to HIV acquisition³⁰. All scenarios included male medical circumcision³¹ at 22% coverage and 60% reduction in acquisition risk.

We configured the EMOD health care system module to follow trends in antiretroviral therapy (ART) expansion in South Africa. Treatment begins with voluntary counseling and testing (VCT), antenatal and infant testing, symptom-driven testing, and low level of couples testing. The model includes loss to follow-up between diagnosis and staging, between staging and linkage to ART or pre-ART care, and during ART or pre-ART care³². Base case projections of South Africa treatment expansion in the reference group are calibrated to reflect a gradual decline of HIV incidence without elimination, so that HIV remains endemic through 2050 (Eaton et al. Lancet Global Health 2014) keeping present-day ART coverage levels of 60%. In a more optimistic scenario ('90-90-90'), we increased testing and linkage to ART, decreased lost-to-follow-up to reflect 90-90-90 UNAIDS goals, such that in the model by 2032 90% of all individuals ever tested positive would be on ART."

6. **Supplement: line 374: I am not sure where the authors obtain the estimates on PrEP efficacy levels of 50%. Does that mean, when a person is on PrEP, the probability of becoming infected is 0.5 times the probability to become infected when not on PrEP? On the cdc webpage <https://www.cdc.gov/hiv/basics/prep.html>, the efficacy is estimated with 70-90%.**

In our model, PrEP efficacy is implemented as reduction in per exposure probability of HIV acquisition. The above mentioned efficacy values from CDC are contingent on perfect adherence, and have been achieved in low-income country settings only in the Partners-PrEP study. In all other studies efficacy was at most 50% (VOICE, iPrEX, FEMPrEP), mainly due to poor adherence (see ALVAC Report 2013, p. 10 and meta-regression analysis in Fonner *et al.*³³). Key evidence about the adherence and thus long-term population-level effectiveness of oral PrEP for key populations in Africa is not yet available. Since long-acting formulations of PrEP could help overcoming the apparent adherence challenges, we stipulated intermediate levels of adherence and thus efficacy (following the meta-regression), and simulated the impact of increased adherence starting from levels observed in clinical trials. This has now been clarified in the *Methods* section (*Supplemental Information I*, Page 16, bottom of the page continuing onto top of Page 17).

Minor points

7. **Please add a table to the supplement with all abbreviations used if the journal allows that.**

We have addressed this by revising and ensuring that the first mention of all abbreviation contains the complete words.

8. **Supplement: line 303 there is one “\” where it does not belong**

We thank the reviewer for pointing out the typo. We have rectified it.

9. Manuscript: line 329 should be “dosage form” instead of “dosage dorm”

We thank the reviewer for pointing out the typo. We have rectified it.

10. Figure 1D: Why are the plasma concentrations this high and constant over time? If you just look at the sum of the individual concentrations, one would not see this constant level. Are the green dots measured values?

Figure 1D is a schematic representing what we would expect an ideal system to look like. The other lines in the graph are not plasma concentrations, but the fraction of drug release from the different formulations we load onto the dosage form; thus a simple additive relationship does not exist. However, we now realize that this might cause confusion for the readers, and we have now highlighted that Figure 1D is a schematic and not actual data.

11. Figure 2ABC and Figure 3B: Do not use histograms when actually representing three independent replicates. Instead it is better practise to show the actual measurements. This is much more meaningful than a mean with standard variation.

We thank the reviewer for highlighting this and agree with them and have updated the manuscript accordingly.

12. Figure 3A, CDE; Figure 4 and Supplement Figure 6: Also in these figures it would be better to show the individual measurements (again n=3) instead of the mean. The reader would obtain a much clearer view on the actual processes and especially in Figure 4 and SuppFig 6 could compare the outcomes in different animals.

We agree with the reviewer and have now represented the data as suggested by the Reviewer.

References

1. Margolis, D. A. *et al.* Cabotegravir plus rilpivirine, once a day, after induction with cabotegravir plus nucleoside reverse transcriptase inhibitors in antiretroviral-naive adults with HIV-1 infection (LATTE): a randomised, phase 2b, dose-ranging trial. *Lancet. Infect. Dis.* **15**, 1145–55 (2015).
2. Rojas, J. *et al.* Dolutegravir monotherapy in HIV-infected patients with sustained viral suppression. *J. Antimicrob. Chemother.* **71**, 1975–81 (2016).

3. Gubavu, C. *et al.* Dolutegravir-based monotherapy or dual therapy maintains a high proportion of viral suppression even in highly experienced HIV-1-infected patients. *J. Antimicrob. Chemother.* **71**, 1046–50 (2016).
4. Capetti, A. F. *et al.* Switch to Dolutegravir plus Rilpivirine Dual Therapy in cART-Experienced Subjects: An Observational Cohort. *PLoS One* **11**, e0164753 (2016).
5. Oldenbuettel, C. *et al.* Dolutegravir monotherapy as treatment de-escalation in HIV-infected adults with virological control: DoluMono cohort results. *Antivir. Ther.* **22**, 169–172 (2017).
6. Rokx, C., Schurink, C. A. M., Boucher, C. A. B. & Rijnders, B. J. A. Dolutegravir as maintenance monotherapy: first experiences in HIV-1 patients. *J. Antimicrob. Chemother.* **71**, 1632–1636 (2016).
7. Radzio, J. *et al.* The long-acting integrase inhibitor GSK744 protects macaques from repeated intravaginal SHIV challenge. *Sci. Transl. Med.* **7**, 270ra5 (2015).
8. Andrews, C. D. *et al.* A long-acting integrase inhibitor protects female macaques from repeated high-dose intravaginal SHIV challenge. *Sci. Transl. Med.* **7**, 270ra4–270ra4 (2015).
9. Markowitz, M. *et al.* Safety and tolerability of long-acting cabotegravir injections in HIV-uninfected men (ECLAIR): a multicentre, double-blind, randomised, placebo-controlled, phase 2a trial. *Lancet HIV* (2017). doi:10.1016/S2352-3018(17)30068-1
10. Verloes, R. *et al.* Safety, tolerability and pharmacokinetics of rilpivirine following administration of a long-acting formulation in healthy volunteers. *HIV Med.* **16**, 477–484 (2015).
11. Boxenbaum, H. Interspecies variation in liver weight, hepatic blood flow, and antipyrine intrinsic clearance: Extrapolation of data to benzodiazepines and phenytoin. *J. Pharmacokinet. Biopharm.* **8**, 165–176 (1980).
12. Frey, B. M., Sieber, M., Mettler, D., Ganger, H. & Frey, F. J. Marked interspecies differences between humans and pigs in cyclosporine and prednisolone disposition. *Drug Metab. Dispos.* **16**, 285–289 (1988).
13. Wurzburger, R. J., Miller, R. L., Boxenbaum, H. G. & Spector, S. Radioimmunoassay of atropine in plasma. *J. Pharmacol. Exp. Ther.* **203**, 435 LP-441 (1977).
14. Iglay, K. *et al.* Systematic Literature Review and Meta-analysis of Medication Adherence With Once-weekly Versus Once-daily Therapy. *Clin. Ther.* **37**, 1813–21.e1 (2015).
15. Kishimoto, H. & Maehara, M. Compliance and persistence with daily, weekly, and monthly bisphosphonates for osteoporosis in Japan: analysis of data from the CISA. *Arch. Osteoporos.* **10**, 27 (2015).
16. Jackson, A. & McGowan, I. Long-acting rilpivirine for HIV prevention. *Curr. Opin. HIV AIDS* **10**, 253–257 (2015).
17. Kovarova, M. *et al.* Nanoformulations of Rilpivirine for Topical Pericoital and Systemic

- Coitus-Independent Administration Efficiently Prevent HIV Transmission. *PLoS Pathog.* **11**, e1005075 (2015).
18. Jackson, A. G. A. *et al.* A compartmental pharmacokinetic evaluation of long-acting rilpivirine in HIV-negative volunteers for pre-exposure prophylaxis. *Clin. Pharmacol. Ther.* **96**, 314–23 (2014).
 19. McAllister, J. W. *et al.* Dolutegravir with tenofovir disoproxil fumarate-emtricitabine as HIV postexposure prophylaxis in gay and bisexual men. *AIDS* **31**, 1291–1295 (2017).
 20. Adams, J. L. *et al.* Single and Multiple Dose Pharmacokinetics of Dolutegravir in the Genital Tract of HIV Negative Women. *Antivir. Ther.* **18**, 1005–1013 (2013).
 21. Greener, B. N. *et al.* Dolutegravir Pharmacokinetics in the Genital Tract and Colorectum of HIV Negative Men After Single and Multiple Dosing. *J. Acquir. Immune Defic. Syndr.* **64**, 39–44 (2013).
 22. Shibata, A. Development of Dolutegravir Combination Nanoparticle Fabrications for HIV Prophylaxis. in *17th International Workshop on Clinical Pharmacology of HIV and Hepatitis Therapy* (17th International Workshop on Clinical Pharmacology of HIV and Hepatitis Therapy, 2016).
 23. Bellinger, A. M. *et al.* Oral, ultra-long-lasting drug delivery: Application toward malaria elimination goals. *Sci. Transl. Med.* **8**, 365ra157-365ra157 (2016).
 24. Bershteyn, A., Klein, D. J. & Eckhoff, P. A. Age-dependent partnering and the HIV transmission chain: a microsimulation analysis. *J. R. Soc. Interface* **10**, (2013).
 25. Ott, M. Q., Bärnighausen, T., Tanser, F., Lurie, M. N. & Newell, M.-L. Age-gaps in sexual partnerships: seeing beyond ‘sugar daddies’. *AIDS* **25**, (2011).
 26. Harling, G., Tanser, F., Mutevedzi, T. & Bärnighausen, T. Assessing the validity of respondents’ reports of their partners’ ages in a rural South African population-based cohort. *BMJ Open* **5**, (2015).
 27. de Oliveira, T. *et al.* Transmission networks and risk of HIV infection in KwaZulu-Natal, South Africa: a community-wide phylogenetic study. *Lancet HIV* **4**, e41–e50 (2017).
 28. Attia, S., Egger, M., Müller, M., Zwahlen, M. & Low, N. Sexual transmission of HIV according to viral load and antiretroviral therapy: systematic review and meta-analysis. *AIDS* **23**, (2009).
 29. Marconi, V. C. *et al.* Cumulative viral load and virologic decay patterns after antiretroviral therapy in HIV-infected subjects influence CD4 recovery and AIDS. *PLoS One* **6**, e17956 (2011).
 30. O’Donnell, M. J. *et al.* Risk factors for ischaemic and intracerebral haemorrhagic stroke in 22 countries (the INTERSTROKE study): a case-control study. *Lancet* **376**, 112–123 (2017).
 31. Connolly, C., Simbayi, L. C., Shanmugam, R. & Nqeketo, A. Male circumcision and its relationship to HIV infection in South Africa: Results of a national survey in 2002 .

SAMJ: South African Medical Journal **98**, 789–794 (2008).

32. Mugglin, C. *et al.* Retention in Care of HIV-Infected Children from HIV Test to Start of Antiretroviral Therapy: Systematic Review. *PLoS One* **8**, e56446 (2013).
33. Fonner, V. A. *et al.* Effectiveness and safety of oral HIV preexposure prophylaxis for all populations. *AIDS* **30**, 1973–1983 (2016).

Reviewers' comments:

Reviewer #1 (Remarks to the Author):

1. The response to Reviewer 1, comment #2, didn't answer the question. The question is whether the investigators tested if salt forms they synthesized have the same pharmacokinetic behavior as those of the manufacturers. It is not always the case, as has been demonstrated for some generic drugs vs. name brand.
2. The response to Reviewer 2, comment #5, needs revision. At this time, the only approved regimen for PrEP is the combination of tenofovir disoproxil fumarate and emtricitabine – a two drug regimen. The efficacy of a single drug regimen has yet to be proven. Thus, the authors comment that “if used for PrEP only one drug is sufficient” isn't established.
3. In the response to Reviewer 3, comment #1, the authors state that they are working on a separate modeling paper. If that is the case, then this does re-raise the concern by Reviewer #1, comment #6, as to whether the modeling should be part of this paper. The authors state the paper in progress will have “more realistic aspects of treatment”, and if that is the case then why include less complete, less realistic modeling in the present work?
4. The authors should include in the main paper (e.g., section: Modelling the impact of long-acting antiretrovirals: population level) that the EMOD-HIV models transmission in heterosexual relationships.
5. In the discussion comments on dolutegravir (DTG) monotherapy, the authors need to acknowledge emerging concerns with that strategy. At the CROI conference earlier this year, the emergence of DTG resistance when used as monotherapy was reported (see: <http://www.croiconference.org/sessions/dolutegravir-maintenance-mono-therapy-hiv-1-randomized-clinical-trial>). Also, see an editorial by Joel Gallant in Antiviral Therapy (doi: 10.3851/IMP3113) and a paper just published from work in humanized mice that DTG monotherapy failed to suppress HIV viremia (doi.org/10.1093/jac/dkx195) .

Reviewer #2 (Remarks to the Author):

The authors have adequately addressed my concerns.

Reviewer #3 (Remarks to the Author):

Kirtane et al addressed all my points raised in my first review in their revised manuscript “Development of an oral once-weekly drug delivery system for HIV anti-retroviral therapy”. My suggestions focused on the modelling part and gladly, the other two reviewers commented on the experimental details of the manuscript.

The authors now describe the model in more detail and communicate the caveats of their model more openly. This was important especially because the results rely on many assumptions on different parameters that have not conclusively determined. The modelling

results support the initial and intuitive hypothesis that the weekly dosage form increases effectiveness of anti-retrovirals both in treatment and prevention. This statement might not change upon the exact model parameters. The authors state in their revised manuscript "We are in the process of preparing a separate manuscript in which many more realistic aspects of treatment will be included in the model, such as inter-patient variation in parameters, more detailed pharmacokinetic models, combination treatment, and pre-exposure prophylaxis." However, I am not a big fan of outsourcing more thorough analyses to another manuscript. The editors as well as the authors themselves need to decide whether this strategy is a valid approach for Nature Communications.

If both parties decide that further analyses of model results are not necessary, I still would like to suggest two changes in the new supplementary material:

(i) The notations in equations 5 and 6 are very non-mathematical: X is defined as total CD4 concentration. Thus the first equation in (5) should start with X , in the following lines X should not appear on the left side of " $=$ ". In addition, the Δ_{CD4} should be defined. The same things should be changed in equation (6).

(ii) The viral strain for which the IC50s in table 3 were measured needs to be mentioned in the figure legend.

Point-by-Point Response

Reference: NCOMMS-17-08538A

We are extremely grateful to the reviewers and the editorial team for their time and consideration of our manuscript. In this document, we address their comments, and we have made changes to the manuscript to reflect their suggestions.

Reviewers' comments:

Reviewer #1 (Remarks to the Author):

- 1. The response to Reviewer 1, comment #2, didn't answer the question. The question is whether the investigators tested if salt forms they synthesized have the same pharmacokinetic behavior as those of the manufacturers. It is not always the case, as has been demonstrated for some generic drugs vs. name brand.**

We apologize for not fully addressing the reviewer's comments in the previous submission. The salt forms of the FDA approved drugs used in our studies are chemically and physically identical to ones described by the companies holding patents on these drugs, and we now provide supporting data here and in the Supplemental Information of the manuscript. However, direct comparison of the pharmacokinetics of our formulation (which includes the active pharmaceutical ingredient i.e. the salt forms of the drugs and excipients) to the brand name is not possible at this time, as the composition of excipients used in the brand name product are not completely known. For example, the FDA label of Tivicay includes information about which excipients are used but not their precise quantities, unlike information about the type of salt form used in the product which is clearly defined¹. Moreover, we aimed in our study to change the pharmacokinetics through the use of gastric resident dosage forms and specific polymer-based excipients. As the drugs used in our immediate release controls and sustained release formulations are the same, it is reasonable to conclude that extended drug concentrations in the plasma (up to 7 days) are achieved due to sustained residence of the dosage form in the stomach and sustained release for 7 days and not due to special salt forms. We have now highlighted this information too in the manuscript for the readers' benefit (Page 16, bottom of the page).

With respect to the chemical and physical forms of the active pharmaceutical ingredient we used, we synthesized the salt forms of the drugs by a procedure similar to ones described in the patents held by the companies commercializing these drugs^{2,3} and hence differences in the salt forms are not expected. We confirm this by using two sets of characterization techniques intended to identify the chemical and physical form of the salt viz. nuclear magnetic resonance (NMR) and X-ray diffraction (XRD) respectively. We compare our results (for dolutegravir sodium and rilpivirine hydrochloride) to those reported by the manufacturers. Such comparisons for cabotegravir sodium are not possible yet, however we report our data.

NMR analysis of our sample of dolutegravir sodium reveals peaks at 10.7, 7.9, 7.4, 7.3, 7.1, 5.2, 4.8, 4.5, 4.4, 4.3, 4.0, 3.8, 1.8, 1.4 and 1.2 ppm. All these peaks are in agreement with the analysis reported by Shionogi & Co. Ltd. and Glaxosmithkline LLC (under 'Example 11'). XRD analysis of our sample of dolutegravir sodium shows peaks at 2θ values of 6.4° , 9.3° , 13.9° ,

19.3° and 21.8° in agreement with the analysis reported by Shionogi & Co. Ltd. and Glaxosmithkline LLC (Claim 16, i). They report peaks at 2Θ values of $6.4^{\circ}\pm 0.2^{\circ}$, $9.2^{\circ}\pm 0.2^{\circ}$, $13.8^{\circ}\pm 0.2^{\circ}$, $19.2^{\circ}\pm 0.2^{\circ}$, $21.8^{\circ}\pm 0.2^{\circ}$. These analyses indicate that both, the chemical identity and the crystal form of dolutegravir sodium used in our studies is as described by the brand name producers.

XRD patterns of the rilpivirine salt also resonate with those described in the patent documents of Janssen Pharmaceutica NV. We observed peaks at 2Θ values of 9.7° , 11.0° , 13.5° , 15.0° , 22.0° and 26.6° and those reported in the patent (Claims 3 and 4) are at $9.7^{\circ}\pm 0.2^{\circ}$, $11^{\circ}\pm 0.2^{\circ}$, $13.5^{\circ}\pm 0.2^{\circ}$, $15^{\circ}\pm 0.2^{\circ}$, $22^{\circ}\pm 0.2^{\circ}$, $26.7^{\circ}\pm 0.2^{\circ}$. NMR peaks for rilpivirine are observed at 10.3, 9.9, 8.1, 7.7-7.6, 6.5, and 2.2 ppm.

The NMR spectrum and XRD pattern of cabotegravir sodium are also described below. We have included all this data in the supplemental information of the revised manuscript (Supplementary Figures 5 and 6).

Figure 1: NMR analysis of antiretroviral drugs

Figure 2: XRD analysis of antiretroviral drugs

- 2. The response to Reviewer 2, comment #5, needs revision. At this time, the only approved regimen for PrEP is the combination of tenofovir disoproxil fumarate and emtricitabine – a two drug regimen. The efficacy of a single drug regimen has yet to be proven. Thus, the authors comment that “if used for PrEP only one drug is sufficient” isn’t established.**

We thank the reviewer for the comment, and in describing capsule size in the revised manuscript, we only assume a minimal of a two-drug regimen.

We have removed the sentence “For PrEP, one drug may be sufficient (hence requiring 2-3 capsules/patient in a single administration event)” from the previous version of our manuscript. This sentence was previously in the bottom half of Page 14. In addition, the manuscript reads “For both treatment and PrEP, two drugs may be required, which could be administered in 5-6 capsules, all administered in a single administration event.” This sentence is in the bottom half of Page 14 and highlighted.

- 3. In the response to Reviewer 3, comment #1, the authors state that they are working on a separate modeling paper. If that is the case, then this does re-raise the concern by Reviewer #1, comment #6, as to whether the modeling should be part of this paper. The authors state the paper in progress will have “more realistic aspects of treatment”, and if that is the case then why include less complete, less realistic modeling in the present work?**

We apologize for our miscommunication about the models included in the present work. As with any model, we are continually seeking to improve their predictive potential through the addition of relationships and parameters brought to light from new or improved data. Our comments on the “more realistic aspects of treatment” in the previous round of reviews were included to indicate that our modeling work is continually evolving as new information is obtained, a standard statement in the modeling field. To be certain, we have included the most realistic and detailed description of HIV dynamics that is available with current data in our manuscript. We believe that the inclusion of both the “within-host” and the “population-level” models situate our work in real-world settings, and that it would be detrimental to the impact of our work to remove them.

- 4. The authors should include in the main paper (e.g., section: Modelling the impact of long-acting antiretrovirals: population level) that the EMOD-HIV models transmission in heterosexual relationships.**

We have added the following comment: “a network transmission model of HIV that includes heterosexual and mother-to-child HIV transmission” to this section “Modelling the impact of long-acting antiretrovirals: population level” on Page 13.

In addition, we have made several stylistic changes to the Supplemental Methods to clarify various aspects of the model.

5. In the discussion comments on dolutegravir (DTG) monotherapy, the authors need to acknowledge emerging concerns with that strategy. At the CROI conference earlier this year, the emergence of DTG resistance when used as monotherapy was reported (see: <http://www.croiconference.org/sessions/dolutegravir-maintenance-monotherapy-hiv-1-randomized-clinical-trial>). Also, see an editorial by Joel Gallant in Antiviral Therapy (doi: 10.3851/IMP3113) and a paper just published from work in humanized mice that DTG monotherapy failed to suppress HIV viremia (doi.org/10.1093/jac/dkx195).

We thank the reviewer for bringing this to our attention. We have now included these references in the discussion section of the manuscript and highlight that “DTG has been shown to be effective as monotherapy or in combination with RPV for treating patients with suppressed viral loads, albeit cases of resistance to DTG monotherapy are emerging and its use as such needs serious consideration.” (Page 14, middle of the page)

Reviewer #2 (Remarks to the Author):

The authors have adequately addressed my concerns.

We thank the reviewer for their time and comments.

Reviewer #3 (Remarks to the Author):

Kirtane et al addressed all my points raised in my first review in their revised manuscript “Development of an oral once-weekly drug delivery system for HIV anti-retroviral therapy”. My suggestions focused on the modelling part and gladly, the other two reviewers commented on the experimental details of the manuscript.

The authors now describe the model in more detail and communicate the caveats of their model more openly. This was important especially because the results rely on many assumptions on different parameters that have not conclusively determined. The modelling results support the initial and intuitive hypothesis that the weekly dosage form increases effectiveness of antiretrovirals both in treatment and prevention. This statement might not change upon the exact model parameters. The authors state in their revised manuscript “We are in the process of preparing a separate manuscript in which many more realistic aspects of treatment will be included in the model, such as inter-patient variation in parameters, more detailed pharmacokinetic models, combination treatment, and pre-exposure prophylaxis.” However, I am not a big fan of outsourcing more thorough analyses to another manuscript. The editors as well as the authors themselves need to decide whether this strategy is a valid approach for Nature Communications.

We thank the reviewer for their enthusiasm for the insight provided by the modeling work and apologize for the miscommunication in our previous response. The within-host model included in this manuscript is the most advanced description of HIV dynamics under treatment that we are able to create with existing data. The comment about (hypothetical) future work was simply meant to highlight that we are continually seeking to improve the predictive potential of our

models through the addition of relationships and parameters brought to light from new or improved data.

We believe that both the “within-host” and “population-level” models represent the state-of-the-art and provide crucial clinical and public health context to this study, and can serve as useful tools for other researchers working on long-acting antiretroviral therapy.

If both parties decide that further analyses of model results are not necessary, I still would like to suggest two changes in the new supplementary material:

- 1. The notations in equations 5 and 6 are very non-mathematical: X is defined as total CD4 concentration. Thus the first equation in (5) should start with X, in the following lines X should not appear on the left side of “=“. In addition, the Δ_{CD4} should be defined. The same things should be changed in equation (6).**

We thank the reviewer for bringing this to our attention. Per the reviewer’s comments, we have now corrected the notation of Equations 5 and 6 in the Supplementary Material (Page 7).

- 2. The viral strain for which the IC50s in table 3 were measured needs to be mentioned in the figure legend.**

All pharmacodynamic parameters are measured on the standard reference HIV-1 strain, NL4-3, with a GFP reporter gene inserted and a deletion in the envelope gene to allow only a single round of infection. Details of the experimental procedures can be found in the references cited in the figure caption and in the text describing these methods. In addition, we now include the following sentence in the caption of Figure 3 (Page 22):

“...Pharmacodynamic values (C_{max} , m) are from the Siliciano laboratory [26,35–37], and are based on a single-round infectivity assay using a GFP-encoding, envelope-deficient version of the standard HIV reference strain NL4-3...”

References

1. FDA. Tivicay- Tablets for oral use. (2013). Available at: https://www.accessdata.fda.gov/drugsatfda_docs/label/2013/204790lbl.pdf. (Accessed: 30th September 2017)
2. Yoshida, H., Taoda, Y. & Johns, B. A. Synthesis of carbamoylpyridone hiv integrase inhibitors and intermediates. (2010).
3. Stevens, P. T. A., Peeters, J., Vandecruys, R. P. G., Stappers, A. E. & Copmans, A. H. Salt of 4[[4-[[4-(2-Cyanoethenyl)-2,6-dimethylphenyl]amino]-2-Pyrimidinyl]amino]benzotrile. (2011).

REVIEWERS' COMMENTS:

Reviewer #1 (Remarks to the Author):

The authors have addressed the questions and recommendations of the last critique. I have no additional comments.